# In Silico Identification of Natural Products and World-Approved Drugs Targeting the KEAP1/NRF2 Pathway Endowed with Potential Antioxidant Profile

**Simone Brogi** [1,2,*,†] **, Ilaria Guarino** [1,†] **, Lorenzo Flori** [1] **, Hajar Sirous** [2] **and Vincenzo Calderone** [1]

[1] Department of Pharmacy, University of Pisa, Via Bonanno 6, 56126 Pisa, Italy; ilaria96guarino@gmail.com (I.G.); lorenzo.flori@farm.unipi.it (L.F.); vincenzo.calderone@unipi.it (V.C.)

[2] Bioinformatics Research Center, School of Pharmacy and Pharmaceutical Sciences, Isfahan University of Medical Sciences, Isfahan 81746-73461, Iran; h_sirous@pharm.mui.ac.ir

[*] Correspondence: simone.brogi@unipi.it; Tel.: +39-050-2219613

[†] These authors contributed equally to this work.

**Abstract:** In this study, we applied a computer-based protocol to identify novel antioxidant agents that can reduce oxidative stress (OxS), which is one of the main hallmarks of several disorders, including cancer, cardiovascular disease, and neurodegenerative disorders. Accordingly, the identification of novel and safe agents, particularly natural products, could represent a valuable strategy to prevent and slow down the cellular damage caused by OxS. Employing two chemical libraries that were properly prepared and enclosing both natural products and world-approved and investigational drugs, we performed a high-throughput docking campaign to identify potential compounds that were able to target the KEAP1 protein. This protein is the main cellular component, along with NRF2, that is involved in the activation of the antioxidant cellular pathway. Furthermore, several post-search filtering approaches were applied to improve the reliability of the computational protocol, such as the evaluation of ligand binding energies and the assessment of the ADMET profile, to provide a final set of compounds that were evaluated by molecular dynamics studies for their binding stability. By following the screening protocol mentioned above, we identified a few undisclosed natural products and drugs that showed great promise as antioxidant agents. Considering the natural products, isoxanthochymol, gingerenone A, and meranzin hydrate showed the best predicted profile for behaving as antioxidant agents, whereas, among the drugs, nedocromil, zopolrestat, and bempedoic acid could be considered for a repurposing approach to identify possible antioxidant agents. In addition, they showed satisfactory ADMET properties with a safe profile, suggesting possible long-term administration. In conclusion, the identified compounds represent a valuable starting point for the identification of novel, safe, and effective antioxidant agents to be employed in cell-based tests and in vivo studies to properly evaluate their action against OxS and the optimal dosage for exerting antioxidant effects.

**Keywords:** KEAP1/NRF2 pathway; in silico screening; natural products; world-approved drugs

## 1. Introduction

Oxidative stress (OxS) is caused by a dramatic imbalance between the production of reactive oxygen species (ROSs) and the enzyme systems that detoxify them [1,2]. This imbalance might harm cells and essential biomolecules, which could have a negative effect on the entire organism [3]. In particular, persistent OxS is actively involved in DNA lesions, protein modifications, and lipid peroxidation, which can promote the activation of cellular responses, including apoptosis, inflammation, and endoplasmic reticulum stress [4–8]. Furthermore, prolonged OxS may hamper biological processes such as mitochondrial function/metabolism and autophagy dysregulation, resulting in a defective immunological response, an acceleration of all pathogenic mechanisms, and a worsening of the symptoms

of several pathological conditions [9–13]. In fact, it has been established that OxS plays a pivotal role in the onset, progression, and effects of several disorders (e.g., cancer [14,15]; heart failure and other cardiovascular diseases [16,17]; depression [18]); chronic and degenerative diseases such as atherosclerosis [19] and chronic fatigue syndrome [20]; dermatological diseases such as atopic dermatitis, psoriasis, vitiligo, and lichen planus [21–24]; and neurodegenerative disorders such as Alzheimer's and Parkinson's diseases [25–27]). Despite definite advancements recently, fighting diseases caused by OxS remains challenging [28].

Interestingly, the development of drugs that specifically target OxS-related pathways by acting as antioxidant agents may be an effective pharmacological strategy for the prevention of several diseases [29,30]. Although there has been much interest in the potential modulation of OxS by supplement and pharmacological therapy, expectations have been stifled by the achievement of questionable outcomes after antioxidant interventions [31]. In fact, different vital signaling and metabolic processes are associated with different concentrations of ROSs. Accordingly, medical research has recently placed a strong focus on locating and specifically modifying ROS enzymatic sources that, in a certain arrangement, cause the illness without affecting ROS physiologic signaling and metabolic activities [32]. One of these pharmacological approaches is the activation of nuclear factor erythroid 2-related factor 2 (NRF2), which results in a positive modulation of the endogenous redox homeostatic system [33]. The Kelch-like ECH-associated protein 1 (KEAP1)/NRF2 signaling pathway includes this protein as one of its constituents. Thus, NRF2 plays a significant role in the OxS response by controlling the transcription of more than 250 genes involved in the regulation of redox metabolism, inflammation, and proteostatic balance [34,35]. Briefly, under physiological conditions, the main NRF2 activity inhibitor KEAP1 continuously targets NRF2 for ubiquitination and proteasomal degradation [36]. In contrast, when cells are exposed to stress-inducing endogenous or exogenous stimuli and electrophiles, KEAP1 is no longer able to target NRF2, which can remain in the cytosol and, after phosphorylation, increase nuclear translocation. In the nucleus, NRF2 is part of a complex transcription machinery that can bind to the antioxidant response element (ARE) in the promoter regions of genes that are involved in the production of cellular antioxidants and detoxifying proteins [37,38]. According to the proposed mechanism of action, the targeting of KEAP1 to interfere with NRF2 recognition can culminate in the release and accumulation of NRF2 in the cytosol, enhancing the cellular antioxidant response. As mentioned, this strategy could be relevant for preventing or slowing the recurrence and progression of several disorders [39–45].

Accordingly, in this paper, we have applied a computer-based procedure, with enclosing molecular docking calculations, molecular dynamics (MD) simulations, an estimation of $\Delta G_{bind}$, and a drug-like profile evaluation to identify potential KEAP1 binders that can occupy the NRF2 binding site, precluding the formation of the KEAP1/NRF2 complex, and allowing the activation of the NRF2 pathway to exert antioxidant effects. In order to consider the NRF2-interacting region on the KEAP1 protein, we used this region as a putative binding site for screening two different libraries: one library contained natural products, and the second database contained world-approved and investigational drugs. Using two separate screening procedures and applying a series of subsequent filters, we identified a few undisclosed molecules (both natural products and drugs) that could be repurposed as antioxidant agents. Figure 1 describes the different steps included in the developed computational protocol. In the next section, a detailed description of each step and the related results are reported.

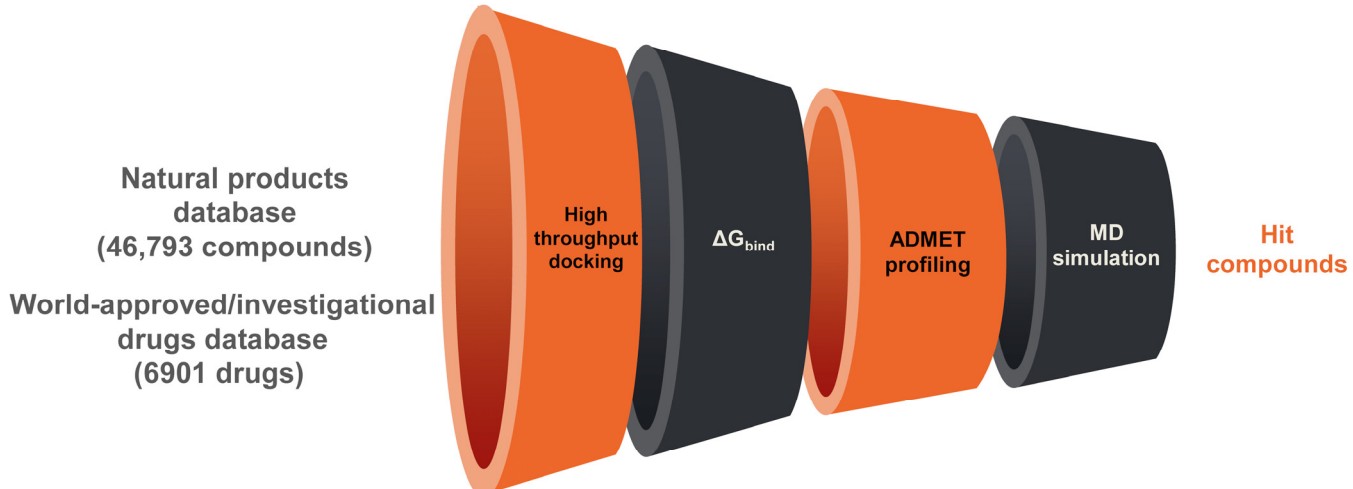

**Figure 1.** The picture illustrates the workflow used in this work for identifying possible antioxidant agents.

## 2. Materials and Methods

### 2.1. Databases and Protein Preparation

The natural product database was prepared by downloading several datasets containing natural products from the ZINC database (https://zinc.docking.org/, accessed on 10 December 2022). In particular, we downloaded chemical structures from the following databases: AfroDb (African medicinal plant), FoodDb, TCM NP (Traditional Chinese medicine), HDMB plant, TIMTEC NP, MolPort NP, IB Screen NP, NPACT, HITNP (herbal ingredients), HIMNP (herbal ingredients in vivo metabolism), Aster NP, SPECS NP, NUBBE natural products, NUTRICHEM, DrugBank Nutraceuticals, Prestwick Phyto NP, DNP natural products, UEFS natural products, TargetMol NP, and INDOFINE natural products. We enclosed all the structures in a unique file, removing redundant structures for a total of 46,793 unique chemical structures.

A database of world-approved drugs was prepared as recently reported [46]. The ZINC database was the source of the world-approved and investigational drugs dataset, which included ~4000 FDA-approved drugs (https://zinc.docking.org/, accessed on 10 December 2022), as well as ~2500 experimental and investigational drugs. In addition, we obtained the Selleckchem dataset of FDA-approved medications, since it had been updated recently (https://www.selleckchem.com/screening/fda-approved-drug-library.html, accessed on 16 June 2023). We extracted 6901 drugs by eliminating redundant chemical structures.

The databases of natural products and approved drugs were prepared using Macro-Model [47] and LigPrep [48] and implemented into Maestro Suite [49], as described [50–52]. In particular, MacroModel was used to minimize the energy of all drugs and natural products, employing OPLS3 as the force field [53]. For non-bonded interactions, a GB/SA solvation model with "no cutoff" was utilized to simulate the solvent effect. A gradient convergence threshold of 0.001 and a maximum of 5000 iterations of the PRCG method were utilized. Subsequently, the compounds were run through the LigPrep program, which produced potential ionization states at a pH of $7.4 \pm 0.2$ and prevented potential mistakes in the respective chemical structures.

The experimentally solved structure of KEAP1 with the NRF2-interacting region was obtained from the protein data bank (PDB ID 2FLU). In addition, we used the three-dimensional structure of the KEAP1 BTB domain (PDB ID 7EXI) (PDB, https://www.rcsb.org/, accessed on 21 April 2023). In particular, the crystal structure 2FLU was used for virtual screening considering the NRF2-interacting region of KEAP1 as the binding site, whereas the crystal structure 7EXI was used in the covalent docking approach because the reactive cysteine residue Cys151 is enclosed in this structure of the BTB domain. The biological assemblies were imported into Maestro and prepared using the Protein Preparation

Wizard implemented in Maestro Suite, as previously communicated [54,55], in order to acquire suitable initial structures for additional computational examination. The substances employed in the crystallization procedure were eliminated.

### 2.2. High-Throughput Docking

For the high-throughput docking process, the Glide software was used, utilizing the Glide standard precision (SP) and extra precision (XP) techniques on the drug database and the protein produced, as previously mentioned [56]. Utilizing a cubic box centered on the NRF2-interacting peptide, energy grids were created with a protein atom scaling factor's default value of 1.0 Å, to cover the entire NRF2 binding site on the KEAP1 Kelch 1 domain [57,58]. Subsequently, using default parameters, the prepared databases were subjected to the molecular docking calculations for the selected binding site. For the post-docking minimization procedure, 50 poses were considered. Evaluation of the Glide SP and XP scores was performed. The main contacts among the natural products and drugs within the designated binding site on the KEAP1 protein were estimated utilizing the application ligand interaction diagram available in the Maestro environment. Likewise, the calculation of the ligand binding energy ($\Delta G_{bind}$), performed by Prime software and utilizing the MM/GBSA technique, was added to the virtual screening procedure [59], as previously described [51,60]. This method combines the solvent-accessible surface area (SASA) for the non-polar solvation term and the continuum solvent generalized Born (GB) model for polar solvation with molecular mechanical (MM) energies. As a result, the Prime procedure was applied to the representative docked poses produced by the molecular docking experiments in order to determine the final ranking of molecules against the chosen enzyme using the MM/GBSA method [61–63]. In order to make the final selection, which took 1% of the compounds in the database into account, a combination of visual inspection (of drugs interacting with important KEAP1 residues involved in the interaction with NRF2) and docking scores, coupled with a satisfactory $\Delta G_{bind}$, was used. We chose compounds with comparable binding modes retrieved from the diverse scoring functions and with SP and XP scores $<-8.00$ kcal/mol for natural products and $<-6.5$ kcal/mol for drugs.

### 2.3. ADMET Profiling

The QikProp application was utilized for calculating the in silico drug-like profiles of the selected molecules [64].

### 2.4. Molecular Dynamics Simulation

Maestro 12.6 was utilized as the graphical interface for setting the MD simulation parameters in the Desmond software, which was used to conduct the MD simulation studies (Desmond Molecular Dynamics System 6.4 academic version, D. E. Shaw Research ("DESRES"), New York, NY, USA, 2020. Maestro-Desmond Interoperability Tools, Schrödinger, New York, NY, USA, 2020). The Desmond software system builder function was utilized to generate three-dimensional ligand/KEAP1 complexes. Using this tool, the ligand/KEAP1 structures were placed into an orthorhombic box and solvated with water molecules represented by the TIP3P water model [65,66]. The MD calculations were conducted utilizing the OPLS force field [53]. The MD simulation studies were accomplished by employing CUDA API technology using two NVIDIA graphics processing units (GPUs). In order to achieve a final salt concentration of 0.15 M, $Na^+$ and $Cl^-$ ions were added to simulate the physiological concentration of monovalent ions. The ensemble class NPT (with a constant number of particles, a pressure of 1.01325 bar, and a temperature of 300 K) was used for the MD simulations. With an inner time step of 2.0 fs, the RESPA integrator was used to calculate the equations, estimating the motion for interactions such as bonded and non-bonded interactions within the short-range cutoff [67]. The Nosé–Hoover thermostat technique was used to maintain a constant temperature throughout the simulation [68], while using the Martyna–Tobias–Klein method, the pressure was maintained constant [69]. Long-range electrostatic interactions were computed using the particle mesh Ewald tech-

nique (PME), whereas the van der Waals and short-range electrostatic interactions were fixed at 9.0 Å [70]. Each system was gradually relaxed and brought to equilibrium using a default procedure that included a number of constrained minimizations and MD simulations. The Desmond package's simulation event analysis tools were utilized to analyze the MD outputs produced during the MD simulation studies. The following equation was applied to determine the Root-Mean-Square Deviation (RMSD) values:

$$RMSD_x = \sqrt{\frac{1}{N}\sum_{i=1}^{N}(r\prime_i(t_x)-r_i(t_{ref}))^2}$$

The RMSDx is utilized to represent the calculation for a frame x; N shows the number of atoms in the system; $t_{ref}$ is used to demonstrate the reference time; and r′ represents the location of the chosen atoms in frame x, after superimposing their position based on the reference frame, where frame x is recorded at time $t_x$.

In addition, the Root-Mean-Square Fluctuation (RMSF) is a crucial statistic parameter utilized to measure the structural dynamics of proteins. By calculating the difference between an atom's or residue's position and its average position over an atomic trajectory, the RMSF is computed by applying the following equation:

$$RMSF_i = \sqrt{\frac{1}{T}\sum_{t=1}^{T}<(r\prime_i(t)-r_i(t_{ref}))^2>}$$

$RMSF_i$ represents the generic residue i; T refers to the trajectory time and the period for which the RMSF is calculated; the reference time is shown by $t_{ref}$; $r_i$ represents the location of the residue i; the exact position of atoms in the residue i after being superposed based on the reference is demonstrated by r′; and the angle brackets represent the average of the square distance, which is taken over based on the selection of atoms in the residue.

*2.5. Covalent Docking*

According to previous reports [60,71,72], covalent docking investigations were conducted utilizing the covalent docking protocol (CovDock) available in Maestro [73]. Prime structure refinement and the Glide docking technique are both used in the algorithm. The CovDock program uses the SMARTS pattern to evaluate customized reactions that are enclosed in a list of potential covalent reactions that are implemented in the program. As a result, the reactive residue and the ligand segment engaged in the reaction can be identified automatically. The reaction involving the right atoms can be written if the desired reaction is not on that list. In this study, because we were interested in exploring different covalent reactions that involved the reactive functional groups of the selected natural products, we selected all the possible covalent reactions listed in the software options. In particular, reactions involving the Michael acceptor and the carbonyl group were indicated as exclusive reactions that can occur considering the selected compounds. The reactive residue of the receptor (Cys151), located in the BTB domain (PDB ID 7EXI), was chosen to begin the calculation, and it was compared with the one specified in the custom chemistry file to determine the type of response. The centroid of the selected residues was used as the grid center, and the grid box dimensions were automatically calculated. To obtain more accurate output results, the pose prediction option was chosen rather than using any constraints. After the docking process, the default settings were used to filter the produced poses, and the MM/GBSA scoring option was selected.

**3. Results and Discussion**

To discover potential antioxidant agents targeting the KEAP1/NRF2 pathway, we performed a high-throughput docking campaign, employing two different chemical libraries (of natural products and world approved and investigational drugs). We considered the NRF2 binding site on the KEAP1 protein to conduct ligand screening using the molecular docking protocol provided in the Glide software. In order to improve the accuracy of our computational procedure, we determined the relative ligand binding energy ($\Delta G_{bind}$).

This method offers a valuable post-scoring filtering strategy to prioritize the screened hit compounds that showed a lower $\Delta G_{bind}$, which was calculated utilizing the MM/GBSA approach. Furthermore, the top-ranked 1% of screened compounds for each library were advanced to the evaluation of physicochemical properties and ADMET profiles. Finally, the compounds with a significant predicted affinity and a satisfactory drug-like profile were selected for further computational analysis. The top-predicted molecules in complex with the KEAP1 protein were utilized as starting points to conduct MD simulation studies for determining their binding stability. The MD simulation experiments were performed utilizing Desmond software, considering 15 complexes (8 KEAP1/natural product complexes and 7 KEAP1/drug complexes) obtained by molecular docking calculations. Because the binding site is on the protein's surface, the MD simulation experiments were essential for this sort of protocol. As a result, it was critical to determine whether a given drug was able to stay in contact with the chosen binding site on the KEAP1 protein or if it might break away from the binding site during the protein dynamic. The following paragraphs provide a detailed presentation of the outcomes of the most promising natural products and the world-approved and investigational drugs in the establishment and preservation of meaningful interactions with the KEAP1 interface involved in the binding with the NRF2.

### 3.1. Natural Products Database Screening

Following the proper preparation of the KEAP1 monomer and the natural products chemical library, which included 46,793 unique chemical structures (more information is provided in the Section 2), we began the high-throughput docking campaign while considering the standard precision (SP) and extra precision (XP) scoring functions available in the Glide software. The top-ranked compounds showing a docking score of −8.00 kcal/mol with the appropriate binding mode were selected; however, no more than 1% of the screened compounds in the docked solution were taken into consideration, a total of 1247 compounds. Furthermore, this subset of selected natural products was subjected to an evaluation of their physicochemical properties to identify suitable drug-like compounds (Table 1). Unfortunately, a large number of natural products did not show satisfactorily predicted physicochemical properties, as expected, and were deprioritized from further investigation. Accordingly, only a few natural products matched the filtering criteria and were then considered for further in silico investigation. In particular, the complexes obtained from the molecular docking studies of the selected natural products were employed as starting points for the MD simulation experiments to evaluate their binding stability. The outputs of the best-performing natural products in establishing and maintaining significant interactions with the KEAP1 interface implicated in the interaction with the NRF2 are presented in detail in the next paragraphs.

**Table 1.** Final hits and their computational parameters obtained from the in silico studies, considering the natural products database.

| Cpd | GlideScore (SP) (kcal/mol) | $\Delta G_{bind}$ (kcal/mol) | SASA [a] | QPlogP [b] | QPlogS [c] | QPPCaco [d] | QPPMDCK [e] | %HOA [f] |
|---|---|---|---|---|---|---|---|---|
| ZINC000000338310 (meranzin hydrate) | −8.254 | −45.94 | 490 | 2.02 | −2.39 | 1505 | 770 | 95 |
| ZINC000059204232 (isoxanthochymol) | −8.698 | −41.96 | 930 | 6.575 | −8.35 | 424 | 197 | 86 |
| ZINC000001531844 (gingerenone A) | −8.426 | −45.23 | 679 | 3.685 | −4.92 | 359 | 165 | 94 |
| ZINC000012153654 (olivil) | −8.223 | −36.62 | 621 | 2.150 | −3.48 | 308 | 139 | 84 |
| ZINC000004095494 (leukotriene A4) | −8.808 | −44.70 | 665 | 5.22 | −4.67 | 289 | 165 | 88 |

| Cpd | GlideScore (SP) (kcal/mol) | ΔG_bind (kcal/mol) | SASA [a] | QPlogP [b] | QPlogS [c] | QPPCaco [d] | QPPMDCK [e] | %HOA [f] |
|---|---|---|---|---|---|---|---|---|
| ZINC000004655402 (5,6-epoxy-8,11,14-eicosatrienoic acid) | −8.197 | −42.27 | 665 | 5.12 | −4.66 | 287 | 164 | 87 |
| ZINC000004655404 (13′-carboxy-γ-tocopherol) | −8.546 | −40.11 | 654 | 5.01 | −4.44 | 277 | 157 | 87 |
| ZINC000004655405 (8,9-epoxyeico-satrienoic acid) | −8.409 | −42.78 | 656 | 4.939 | −4.49 | 254 | 143 | 100 |

[a] SASA predicted the total solvent accessible surface (range or recommended value for 95% of known drugs 300–1000). [b] QPlogP predicted octanol/water partition coefficient (range or recommended value for 95% of known drugs −2–6.5). [c] QPlogS predicted aqueous solubility in mol/dm3 (range or recommended value for 95% of known drugs −6.5–0.5). [d] QPPCaco predicted apparent Caco-2 cell permeability in nm/s (range or recommended value for 95% of known drugs <100 poor, >500 great). [e] QPPMDCK predicted apparent MDCK cell permeability in nm/sec (range or recommended value for 95% of known drugs <100 poor, >500 great). [f] %HOA predicted human oral absorption on 0 to 100% scale (range or recommended value for 95% of known drugs >80% high). Recommended values are reported in the QikProp user manual.

All the trajectories derived from the MD simulation studies were examined to select only those compounds that were able to establish fruitful interactions with the KEAP1 binding site, preserving the binding with the considered interacting region on the target protein. All trajectories were analyzed by calculating their RMSD and RMSF and assessing the dynamic ligand interaction diagrams, combined with a visual inspection of each trajectory. The outputs of the MD simulation experiments are shown in Figure 2. In general, the results of the MD simulations showed that half of the selected natural products in complex with KEAP1 presented a small RMSD along with limited fluctuations of the protein, as indicated by the RMSF, whereas the other natural products showed limited ligand stability with large conformational changes, as indicated by calculating the ligand RMSD. This did not allow stable interactions with the selected binding site, probably highlighting an uncertain affinity for the selected binding site.

The most promising natural products that could target KEAP1, precluding the possibility of interaction with the NRF2 and thus exerting antioxidant effects, are presented in detail in the following sections.

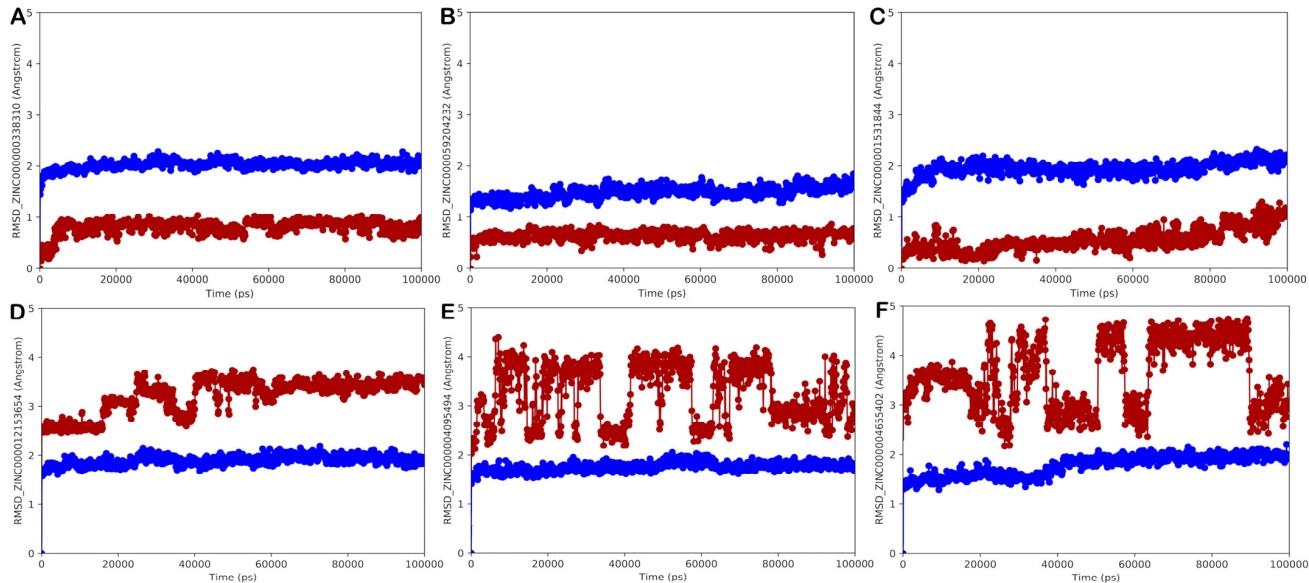

**Figure 2.** *Cont.*

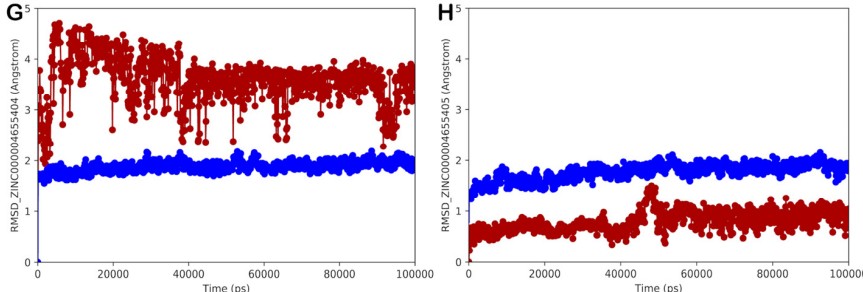

**Figure 2.** RMSD calculation (protein: blue line; and ligand: red line) for each complex (KEAP1/natural product), selected by docking studies, after 100 ns of MD simulation ((**A**) ZINC000000338310; (**B**) ZINC000059204232; (**C**) ZINC000001531844; (**D**) ZINC000012153654; (**E**) ZINC000004095494; (**F**) ZINC000004655402; (**G**) ZINC000004655404; (**H**) ZINC000004655405). Maestro was utilized for generating the pictures.

3.1.1. Potential Natural Products Hit Molecules Targeting the NRF2 Binding Site on KEAP1 Protein

ZINC000000338310 (Meranzin Hydrate)

The compound ZINC000000338310 (meranzin hydrate) is one of the natural products selected by the developed in silico protocol and shows the greatest promise. The molecular docking result is displayed in Figure 3, which highlights the main interactions that this naturally occurring chemical can establish with the KEAP1 binding site (Figure 3A). This molecule was capable of significantly targeting the KEAP1 binding site, thereby establishing a strong network of H-bonds with Ser363, Arg380, Asn382, and Asn414 (Figure 3B). The natural product ZINC000000338310 could occupy the entire surface of the KEAP1 binding site involved in the NRF2 interaction, as depicted in the graphic representation, targeting crucial residues in the formation of the KEAP1/NRF2 complex. This may interfere with the correct recognition of the two proteins, precluding the anchoring of the NRF2 to the KEAP1.

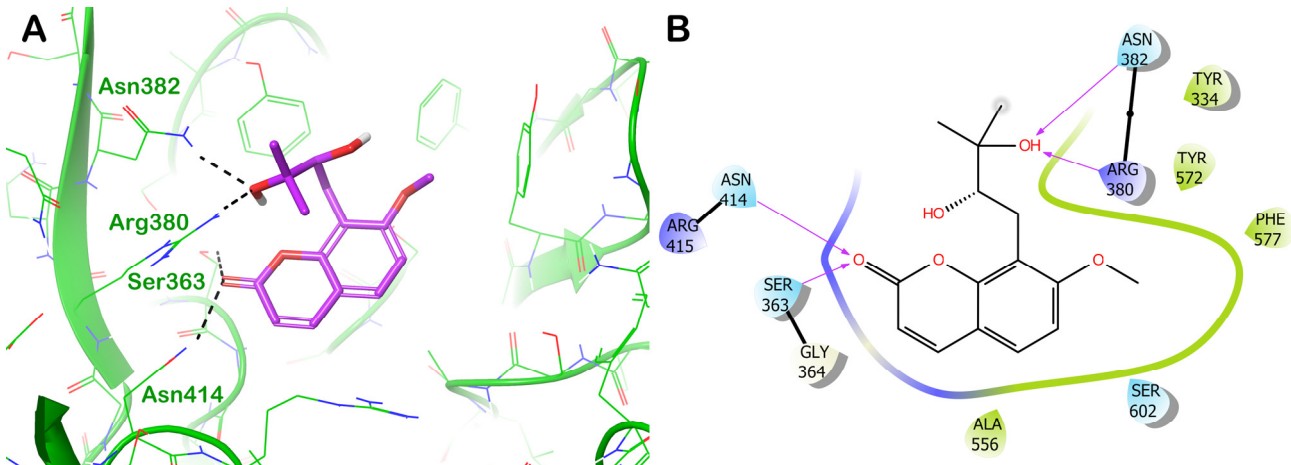

**Figure 3.** (**A**) Binding mode of ZINC000000338310 (light purple sticks) within the selected KEAP1 binding site (PDB ID 2FLU, green cartoon). The interacting amino acids are represented by lines. The grey dotted lines represent the H-bonds. For the sake of clarity, non-polar hydrogens were removed. (**B**) Two-dimensional representation of the contacts established by ZINC000000338310 within the KEAP1 binding site. Maestro and Ligand Interaction Diagram applications were used for generating the pictures.

As previously mentioned, we studied the binding stability using MD simulation experiments in an explicit solvent to improve the screening quality. Figure 4 displays the results of the RMSD and RMSF calculations for ZINC000000338310, as well as trajectory

analysis of the binding stability. During the MD simulation, ZINC000000338310 maintained its binding to the KEAP1 binding site while maintaining the key contacts identified by the molecular docking calculation. In particular, the H-bonds with the residues Ser363, Arg380, Asn382, and Asn414 were well-maintained, becoming water-mediated. Significant hydrophobic interactions were detected with the residues Tyr334, Ala556, Tyr572, and Phe577. In addition, we observed the formation of a more favorable H-bond with Ser602. The docking score (of −8.254 kcal/mol) along with a satisfactory $\Delta G_{bind}$ confirmed the ability of ZINC000000338310 to establish relevant interactions with the KEAP1 surface implicated in the interaction with the NRF2.

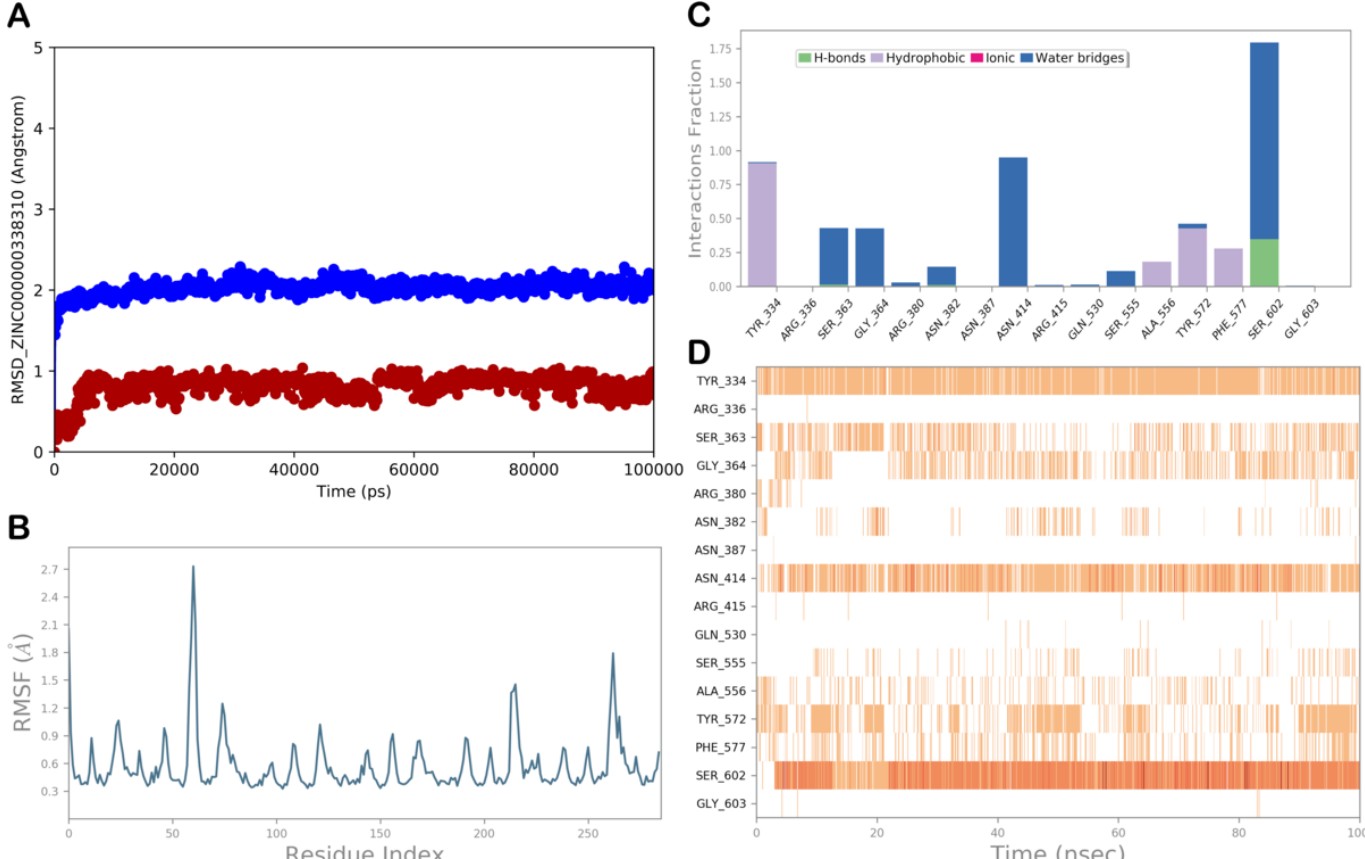

**Figure 4.** (**A**) RMSD evaluation (protein: blue line; and ligand: red line). (**B**) RMSF assessment for the complex KEAP1/ZINC000000338310, obtained by docking studies, following a 100 ns MD simulation. (**C,D**) ZINC000000338310 observed throughout the MD run. Four types of interactions can be distinguished: water bridges (blue), ionic (magenta), hydrophobic (grey), and H-bonds (green). Over the trajectory, the stacked bar charts are normalized. For instance, a value of 0.7 indicates that a particular contact is maintained 70% of the time during simulation. Values greater than 1.0 could occur because a protein residue could interact with the ligand more than once using the same subtype. A timeline explanation of the primary interactions is shown in the following diagram in the figure. Those residues that interact with the ligand in each trajectory frame are displayed in the output. A darker orange hue denotes several contacts that some residues have with the ligand. Maestro and Desmond software tools were utilized to generate the pictures (Maestro, Schrödinger LLC, release March 2020).

ZINC000059204232 (Isoxanthochymol)

Another interesting natural product potentially able to interact with KEAP1, forming stable contacts, is the compound ZINC000059204232 (isoxanthochymol). Based on the docking output, as depicted in Figure 5, the main interactions that the natural compound can establish with the surface of the KEAP1 involved in the interaction with the NRF2

are highlighted (Figure 5A). The selected natural products strongly targeted the residues Arg415 and Ser508 by H-bonds, whereas hydrophobic interactions were found with Tyr334 and Tyr525 (Figure 5B). Considering the relevant residues involved in the interaction with the NRF2 targeted by the studied compound, we hypothesize that ZINC000059204232 interferes with the proper recognition of the two proteins and prevents the NRF2 from anchoring to the KEAP1.

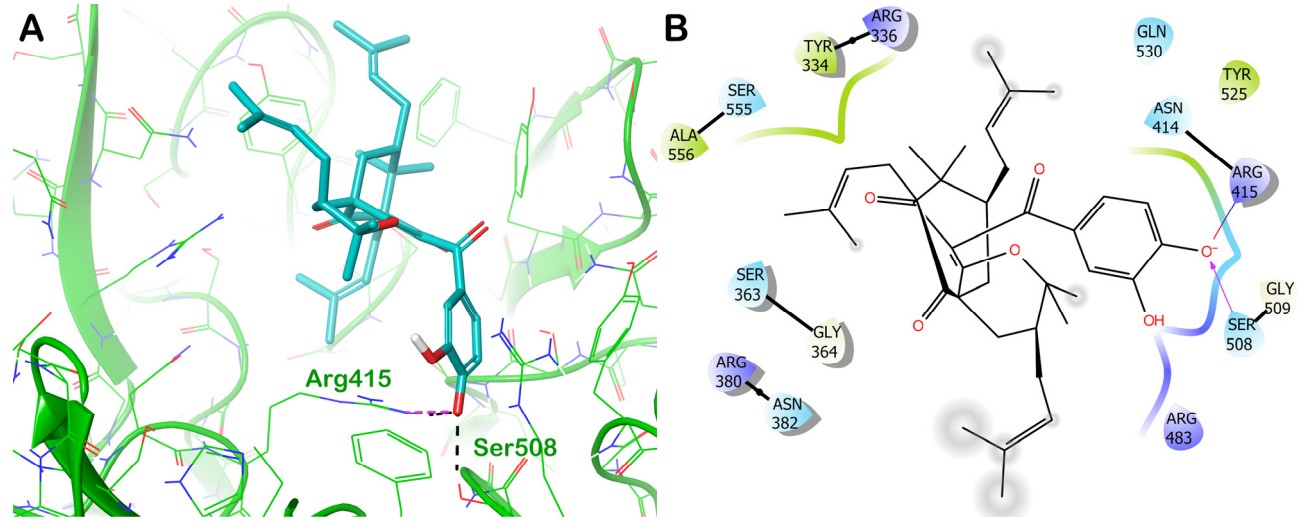

**Figure 5.** (**A**) Binding mode of ZINC000059204232 (cyan sticks) within the selected KEAP1 binding site (PDB ID 2FLU, green cartoon). The interacting amino acids are represented by lines. The grey dotted lines represent the H-bonds. For the sake of clarity, non-polar hydrogens were removed. (**B**) Two-dimensional representation of the contacts established by ZINC000059204232 within the KEAP1 binding site. Maestro and Ligand Interaction Diagram applications were used for generating the pictures.

To confirm the findings of the molecular docking studies, we conducted an MD simulation experiment on the complex KEAP1/ZINC000059204232 (Figure 6). Calculations of the RMSD and RMSF values showed a strong stability of the complex with very small fluctuations in the protein. Analysis of the MD trajectory revealed that the interactions found by the molecular docking calculations were maintained during the MD simulation, including polar interactions with Arg415 and Ser508 and hydrophobic interactions with Tyr334 and Tyr525. We observed further contacts with the residues Arg483, Gln530, Ser555, and Ser602 via water-mediated H-bonds and with Tyr572 and Phe577 via hydrophobic interactions. This binding mode, along with the computational scores, supported the ability of ZINC000059204232 to target KEAP1, positively modulating the NRF2 antioxidant pathway.

ZINC000001531844 (Gingerenone A)

The third interesting natural product identified using the high-throughput docking technique is ZINC000001531844 (gingerenone A). This compound established a remarkable H-bond network with the residues Ser363, Asn382, Arg415, and Ser508, while a π–π stacking was observed with the residue Tyr334 (Figure 7).

To validate the docking output, we inspected the MD simulation trajectory to observe whether the molecule could preserve the binding mode found in molecular docking studies. The MD simulation analysis of ZINC000001531844 is shown in Figure 8. The complex KEAP1/ZINC000001531844 was stable during the MD simulation, as suggested by the RMSD and RMSF values. Upon a thorough examination of the entire MD trajectory, we noticed that the main interactions identified using molecular docking were nicely maintained throughout the simulation. Indeed, the molecule was capable of retaining significant interactions with Tyr334, Ser363, and Arg415, while the H-bonds with Asn382 and Ser508 became mainly sporadic. Interestingly, the natural product ZINC000001531844

could establish additional contacts with Arg380, Asn414, Tyr525, Tyr572, and Ser602. This can contribute to the stabilization of the retrieved binding mode, perfectly anchoring this natural product to the selected binding site and highlighting its potential for interfering with the formation of the KEAP1/NRF2 complex.

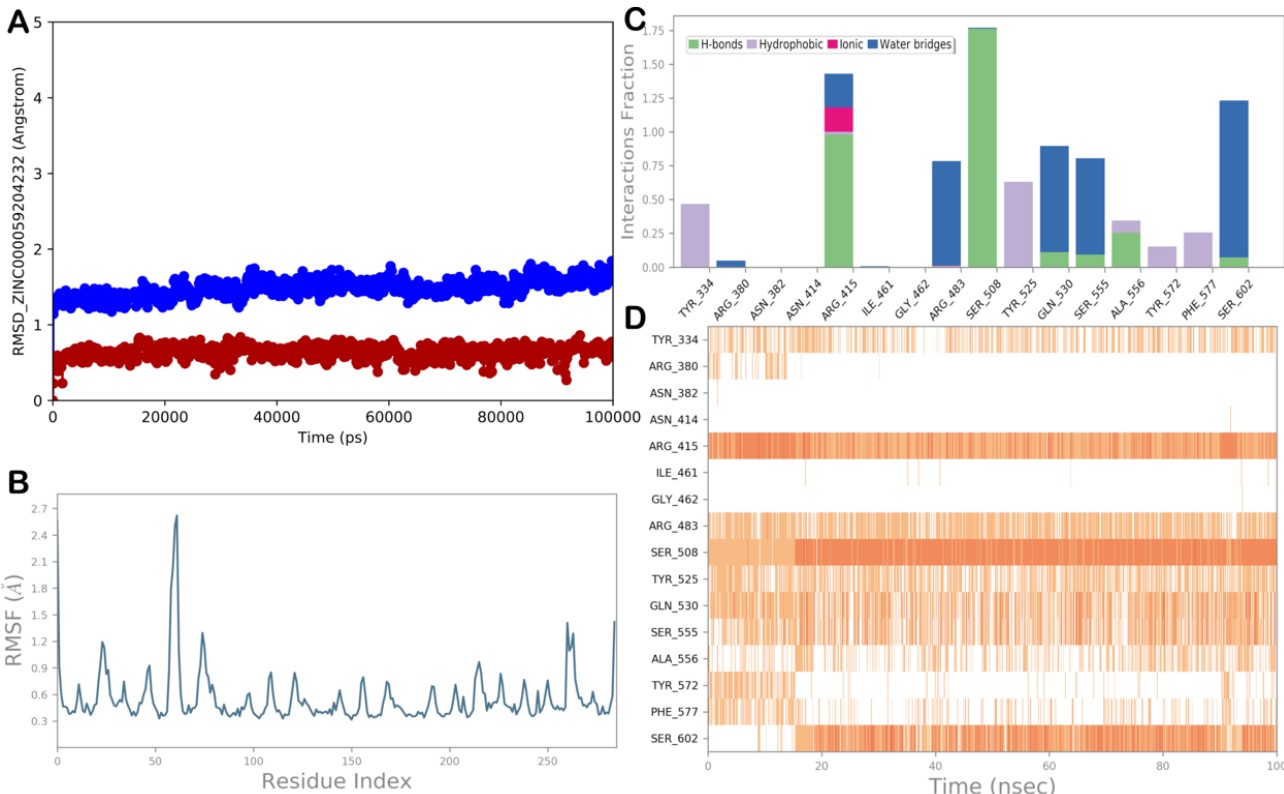

**Figure 6.** (**A**) RMSD evaluation (protein: blue line; and ligand: red line). (**B**) RMSF assessment for the complex KEAP1/ZINC000059204232, obtained by docking studies, following a 100 ns MD simulation. (**C,D**) ZINC000059204232 observed throughout the MD run. Four types of interactions can be distinguished: water bridges (blue), ionic (magenta), hydrophobic (grey), and H-bonds (green). Over the trajectory, the stacked bar charts are normalized. For instance, a value of 0.7 indicates that a particular contact is maintained 70% of the time during simulation. Values greater than 1.0 could occur because a protein residue could interact with the ligand more than once using the same subtype. A timeline explanation of the primary interactions is shown in the following diagram in the figure. Those residues that interact with the ligand in each trajectory frame are displayed in the output. A darker orange hue denotes several contacts that some residues have with the ligand. Maestro and Desmond software tools were utilized to generate the pictures (Maestro, Schrödinger LLC, release March 2020).

To add other information about the undisclosed natural products identified as antioxidant agents, we performed a careful search of the identified compounds to investigate their potential preclinical characterization. The first promising natural product, the compound ZINC000000338310, which is meranzin hydrate (8-[(2S)-2,3-dihydroxy-3-methylbutyl]-7-methoxychromen-2-one), belongs to *Citrus reticulata* Blanco (the common mandarin fruit). Several studies have proven that it possesses antidepressant, antibacterial, and anti-atherosclerosis effects [74–77]. *Citrus reticulata* Blanco has been demonstrated to exert antioxidant effects; however, to the best of our knowledge, none of its components have been identified to possess this effect [78]. Accordingly, meranzin hydrate should be further investigated for its antioxidant profile, with the expectation of being employed as a nutraceutical to supplement dietary regimens to exploit its antioxidant potential. Another interesting natural product is the compound ZINC000059204232,

which is isoxanthochymol ((1*R*,3*R*,9*S*,11*S*)-7-(3,4-dihydroxybenzoyl)-4,4,10,10-tetramethyl-3,9,11-tris(3-methylbut-2-enyl)-5-oxatricyclo [7.3.1.0$^{1,6}$]tridec-6-ene-8,13-dione) and is a polyisoprenylated benzophenone found in different species of *Garcinia* (i.e., fruit rinds, stem bark, seed pericarps, in the leaves of *Garcinia indica*, and in the fruit rinds of *Garcinia cambogia*) [79]. It has been demonstrated to have antitumor and antibacterial activity in the micromolar range [79–81]. Recently, extracts of *Garcinia celebica,* also containing isoxanthochymol, have demonstrated antioxidant effects, although the mechanism of action was not elucidated [82]. Accordingly, in this work, we hypothesized that the antioxidant effects of isoxanthochymol could be ascribed to the targeting of the KEAP1/NRF2 pathway, highlighting this compound as a possible supplement to prevent cellular OxS. Finally, the compound ZINC000001531844, which is gingerenone A ((*E*)-1,7-bis(4-hydroxy-3-methoxyphenyl)hept-4-en-3-one), a linear diarylheptanoid found in the herb *Zingiber officinalis*, was investigated. It showed senolytic properties, promoting the death of senescent cells with no effect on non-senescent cells. These characteristics strongly support the idea that this natural compound may have therapeutic benefits in diseases characterized by senescent cell accumulation [83]. Furthermore, it showed anti-obesity properties, suppressing obesity and inflammation of adipose tissue in mice that were fed a high-fat diet and modulating the metabolism of fatty acids through the in vitro and in vivo activation of the AMP-activated protein kinase (AMPK). In addition, gingerenone A suppressed adipose tissue inflammation by inhibiting macrophage recruitment and downregulating proinflammatory cytokines [84]. It has been shown that the selected compound promoted the antiproliferation and senescence of breast cancer cells induced by OxS [85]. Gingerenone A showed an antibacterial profile, inhibiting the *Staphylococcus aureus* SaHPPK enzyme [86]. Finally, it was reported recently that this natural product relieved ferroptosis in secondary liver injury by activating the NRF2 signaling pathway, but the mechanism of action remains unclear [87]. Accordingly, the activity of gingerenone A could be related to the possibility of targeting KEAP1, increasing the levels of NRF2, and thereby exerting antioxidant effects and modulating several cellular pathways. Therefore, it can be further investigated as a dietary supplement to prevent OxS-induced disorders.

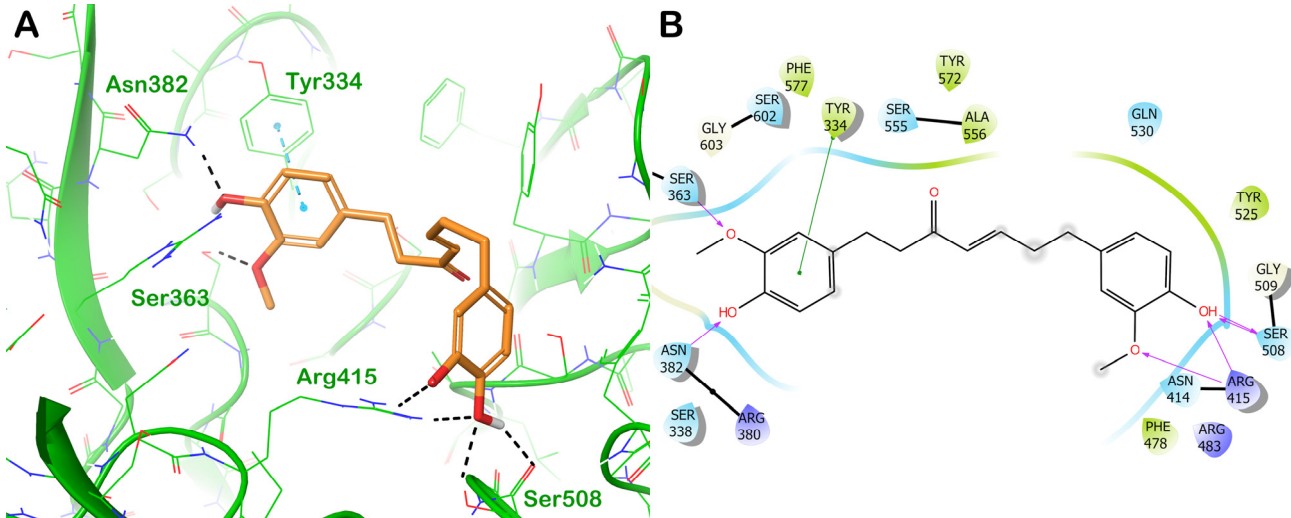

**Figure 7.** (**A**) Binding mode of ZINC000001531844 (orange sticks) within the selected KEAP1 binding site (PDB ID 2FLU, green cartoon). The interacting amino acids are represented by lines. The grey dotted lines represent the H-bonds. For the sake of clarity, non-polar hydrogens were removed. (**B**) Two-dimensional representation of the contacts established by ZINC000001531844 within the KEAP1 binding site. Maestro and Ligand Interaction Diagram applications were used for generating the pictures.

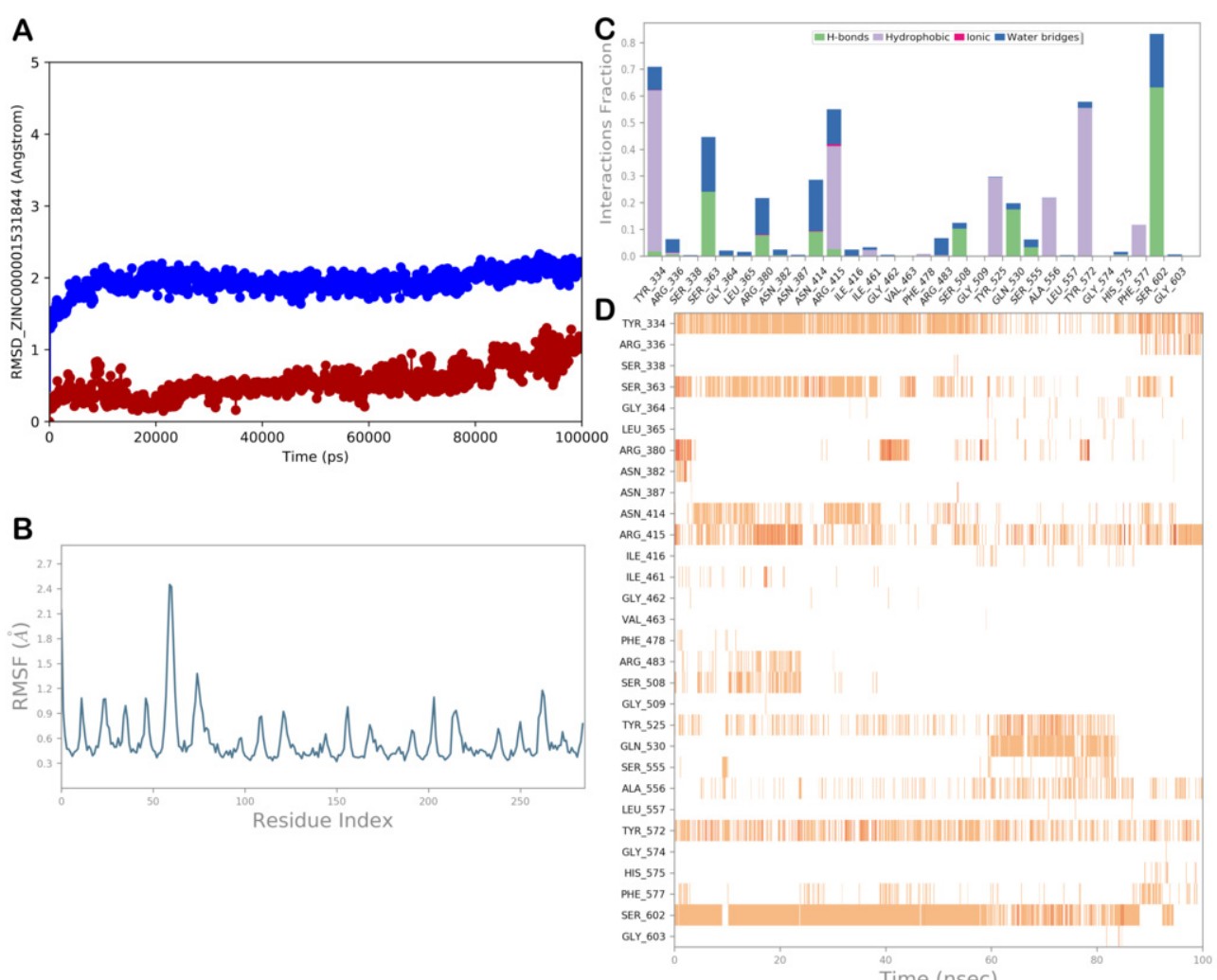

**Figure 8.** (**A**) RMSD evaluation (protein: blue line; and ligand: red line). (**B**) RMSF assessment for the complex KEAP1/ZINC000001531844, obtained by docking studies, following a 100 ns MD simulation. (**C,D**) ZINC000001531844 observed throughout the MD run. Four types of interactions can be distinguished: water bridges (blue), ionic (magenta), hydrophobic (grey), and H-bonds (green). Over the trajectory, the stacked bar charts are normalized. For instance, a value of 0.7 indicates that a particular contact is maintained 70% of the time during simulation. Values greater than 1.0 could occur because a protein residue could interact with the ligand more than once using the same subtype. A timeline explanation of the primary interactions is shown in the following diagram in the figure. Those residues that interact with the ligand in each trajectory frame are displayed in the output. A darker orange hue denotes several contacts that some residues have with the ligand. Maestro and Desmond software tools were utilized to generate the pictures (Maestro, Schrödinger LLC, release March 2020).

### 3.2. Approved and Investigational Drugs Database Screening

Following proper preparation of the KEAP1 protein along with the world-approved and investigational drugs database (including 6901 drugs), as detailed in the Section 2, we started the high-throughput docking campaign considering the scoring functions available in the Glide software. After the procedure, we investigated the therapeutic indication of the retrieved drugs to select only those that can be employed in chronic treatments devoid of serious and severe undesired effects and showing no toxicity against human organisms with high $LD_{50}$ values. In addition, we excluded from further investigation cytotoxic drugs, such as antitumor agents; antibiotic agents, because of the possibility of developing resistance phenomena; anti-inflammatory drugs, because they can cause

gastrointestinal complications after prolonged use; and peptide drugs, because of their rapid metabolism in the human body after oral administration. Considering these filters along with a satisfactory docking score, $\Delta G_{bind}$, and the appropriate binding mode, we selected seven drugs that were able to establish significant interactions with the KEAP1 protein to advance in the MD simulation studies (Table 2).

**Table 2.** Final hits and their computational parameters obtained from in silico studies, considering the approved and investigational drugs database.

| Cpd | GlideScore (SP) (kcal/mol) | $\Delta G_{bind}$ (kcal/mol) | LD$_{50}$ [a] mg/kg | Therapeutic Indications |
|---|---|---|---|---|
| ZINC000003782807 (nedocromil) | −8.243 | −54.91 | 980 | Approved anti-asthma medication. It is used prophylactically in asthma including allergy-related asthma. Ophthalmic nedocromil is used to treat itchy eyes caused by allergies. |
| ZINC000001536201 (sacubitrilat) | −8.089 | −49.77 | 2000 | Active form of sacubitril, and it belongs to the class of therapeutics called angiotensin receptor neprilysin inhibitors. This drug improves endothelial cell function, and it is recommended for treating cardiovascular disorders. |
| ZINC000000538557 (zopolrestat) | −7.148 | −47.53 | 1034 | Zopolrestat is a potent inhibitor of aldose reductase, and it is approved for the treatment of diabetic complications such as diabetic cardiovascular autonomic neuropathy or diabetic neuropathy. |
| ZINC000003948738 (bempedoic acid) | −6.792 | −50.96 | >1000 | Bempedoic acid is a prescription-only, once-daily oral tablet used to lower low-density lipoprotein cholesterol (LDL-C) levels in the blood. It is used for the treatment of hypercholesterolemia. |
| ZINC000003831151 (montelukast) | −6.695 | −41.48 | 1552 | FDA-approved drug for treating chronic asthma and prophylaxis and prevention of exercise-induced bronchoconstriction. It is also approved to relieve seasonal and perennial allergic rhinitis symptoms. |
| ZINC000001547346 (solabegron) | −6.658 | −54.89 | 1036 | Solabegron is a selective adrenergic β-3 adrenoceptor agonist, and it was developed for the treatment of overactive bladder and irritable bowel syndrome. |
| ZINC000013541362 (dinoprost) | −6.531 | −51.32 | 1170 | Dinoprost is a medication used to induce a second trimester abortion and is used in evacuating the uterus in cases of fetal death. It has been investigated for the treatment of headaches. |

[a] LD$_{50}$ values were provided by DrugBank (https://go.drugbank.com/, accessed on 22 July 2023).

To select only those medications capable of forming fruitful interactions with the KEAP1 binding site while maintaining binding with the target protein's regarded interacting area, all MD trajectories were examined. In addition to a visual inspection of each trajectory, all trajectories were evaluated by calculating the RMSD and RMSF and by assessing the dynamic ligand interaction diagrams. The results of the MD simulation experiments are shown in Figure 9. In general, the results of the MD simulations showed that five of the selected drugs in complex with KEAP1 presented a limited RMSD along with slight protein fluctuations, as indicated by the RMSF, whereas the other drugs showed limited ligand

stability with large conformational changes, as determined by calculating the ligand RMSD, which did not allow stable contacts with the selected binding site, probably highlighting an uncertain affinity for the selected binding site.

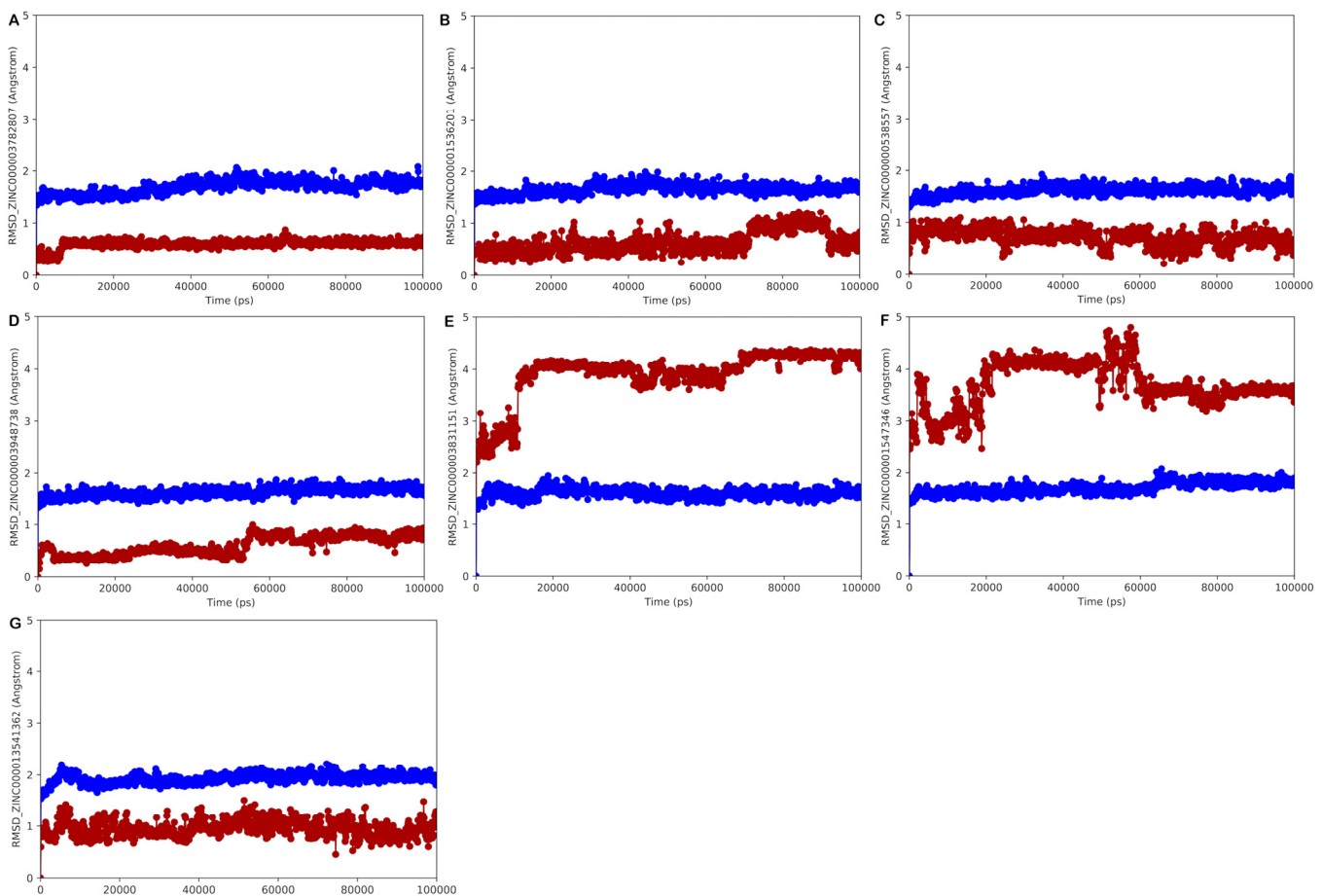

**Figure 9.** RMSD calculation (protein: blue line; and ligand: red line) for each complex (KEAP1/natural product), selected by docking studies, after 100 ns of MD simulation ((**A**) ZINC000003782807; (**B**) ZINC000001536201; (**C**) ZINC000000538557; (**D**) ZINC000003948738; (**E**) ZINC000003831151; (**F**) ZINC000001547346; (**G**) ZINC000013541362). Pictures were generated by Maestro.

Accordingly, the most promising drugs that could target KEAP1, precluding the possibility of interaction with the NRF2 and thus exerting antioxidant effects, are presented in detail in the following sections.

### 3.2.1. Potential Approved and Investigational Drug Hits Targeting the NRF2 Binding Site on KEAP1 Protein

ZINC000003782807 (Nedocromil)

Considering the ligand screening results using world-approved and investigational drugs and applying the mentioned post-search filtering criteria, the drug ZINC000003782807 (nedocromil) is one of the most promising molecules with potential antioxidant effects. In particular, the drug was found to interact with the KEAP1, targeting the NRF2 interacting site. Figure 10 shows the output of the molecular docking calculation. The selected drug could establish several contacts with the active residues of the KEAP1 active site. In particular, we observed a series of polar contacts (with H-bonds and slat bridges) with the residues Arg380, Asn382, Arg415, Arg483, Ser508, Ser555, and Ser602. Moreover, we detected a cation–π stacking with Arg415. Notably, the drug ZINC000003782807 fully occupied the KEAP1 region involved in NRF2 recognition, targeting all crucial residues implicated in the

formation of the KEAP1/NRF2 complex. This could interfere with the correct recognition of the two proteins, preventing the NRF2 from anchoring to the KEAP1.

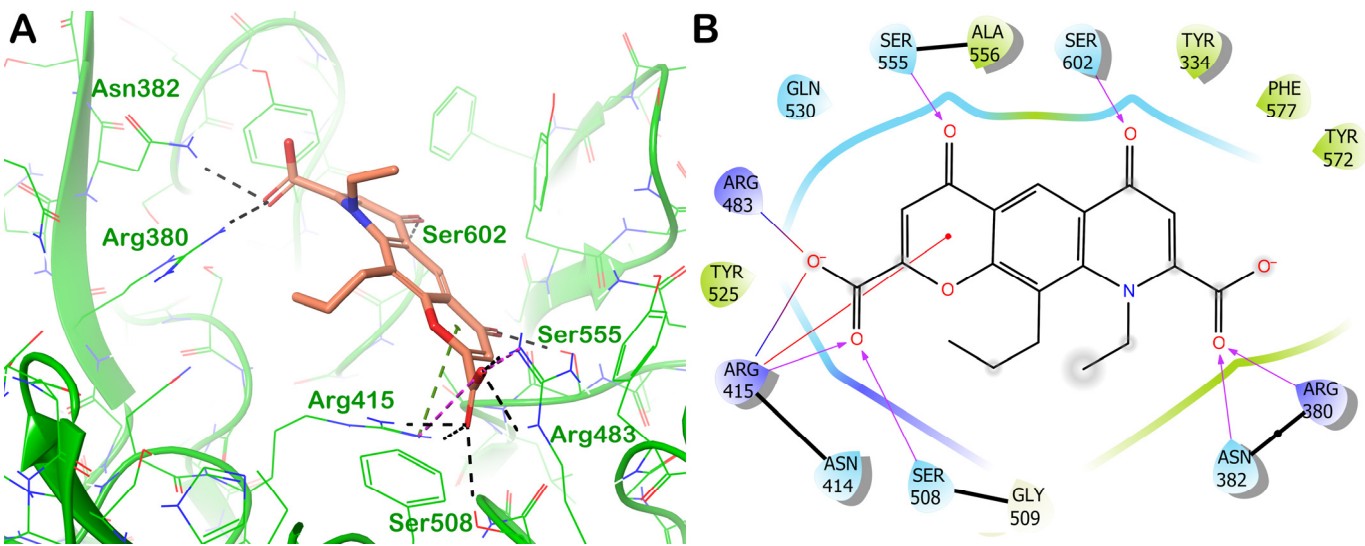

**Figure 10.** (**A**) Binding mode of ZINC000003782807 (pink sticks) within the selected KEAP1 binding site (PDB ID 2FLU, green cartoon). The interacting amino acids are represented by lines. The grey dotted lines represent the H-bonds. For the sake of clarity, non-polar hydrogens were removed. (**B**) Two-dimensional representation of the contacts established by ZINC000003782807 within the KEAP1 binding site. Maestro and Ligand Interaction Diagram applications were used for generating the pictures.

To verify the molecular docking result, we explored the MD simulation path to determine whether the compound could maintain the binding mode observed through the molecular docking investigation. Figure 11 reports the MD simulation analysis related to ZINC000003782807. The MD simulation revealed that the complex KEAP1/ZINC000003782-807 was stable, as indicated by the RMSD and RMSF values. After conducting a thorough analysis of the entire MD trajectory, we found that the most important interactions identified by molecular docking were largely maintained during the MD simulation. In fact, the compound was able to maintain strong contacts with Arg380, Arg415, Arg483, Ser508, Ser555, and Ser602, whereas the H-bond with Asn382 was no longer detectable after the MD simulation was initiated. Furthermore, the strong hydrophobic interaction (cation–π) with Arg415 was well-maintained during the simulation. In addition, water-mediated H-bonds were established by the drug and the backbone of residues Gly384 and Ile416 and can contribute to stabilizing the identified binding mode. This pattern of interaction was a determinant of keeping ZINC000003782807 anchored to the KEAP1 binding site, highlighting its potential for interfering with the formation of the KEAP1/NRF2 complex.

ZINC000000538557 (Zopolrestat)

ZINC000000538557 (zopolrestat) is another intriguing drug that may interact with KEAP1, thereby establishing stable contacts. Figure 12 shows the main interactions that the drug can establish with the surface of the KEAP1, which is involved in the interaction with the NRF2, based on the docking output (Figure 12A). The selected drug can strongly target the residues Arg415, Arg483, and Ser508 by H-bonds and salt bridges, whereas hydrophobic interactions (cation–π stacking) were detected with Arg415 (Figure 12B). Considering the important residues targeted by the compound in its interaction with the NRF2, we postulate that ZINC000000538557 may interfere with the correct recognition of the two proteins, thereby precluding the formation of the KEAP1/NRF2 complex.

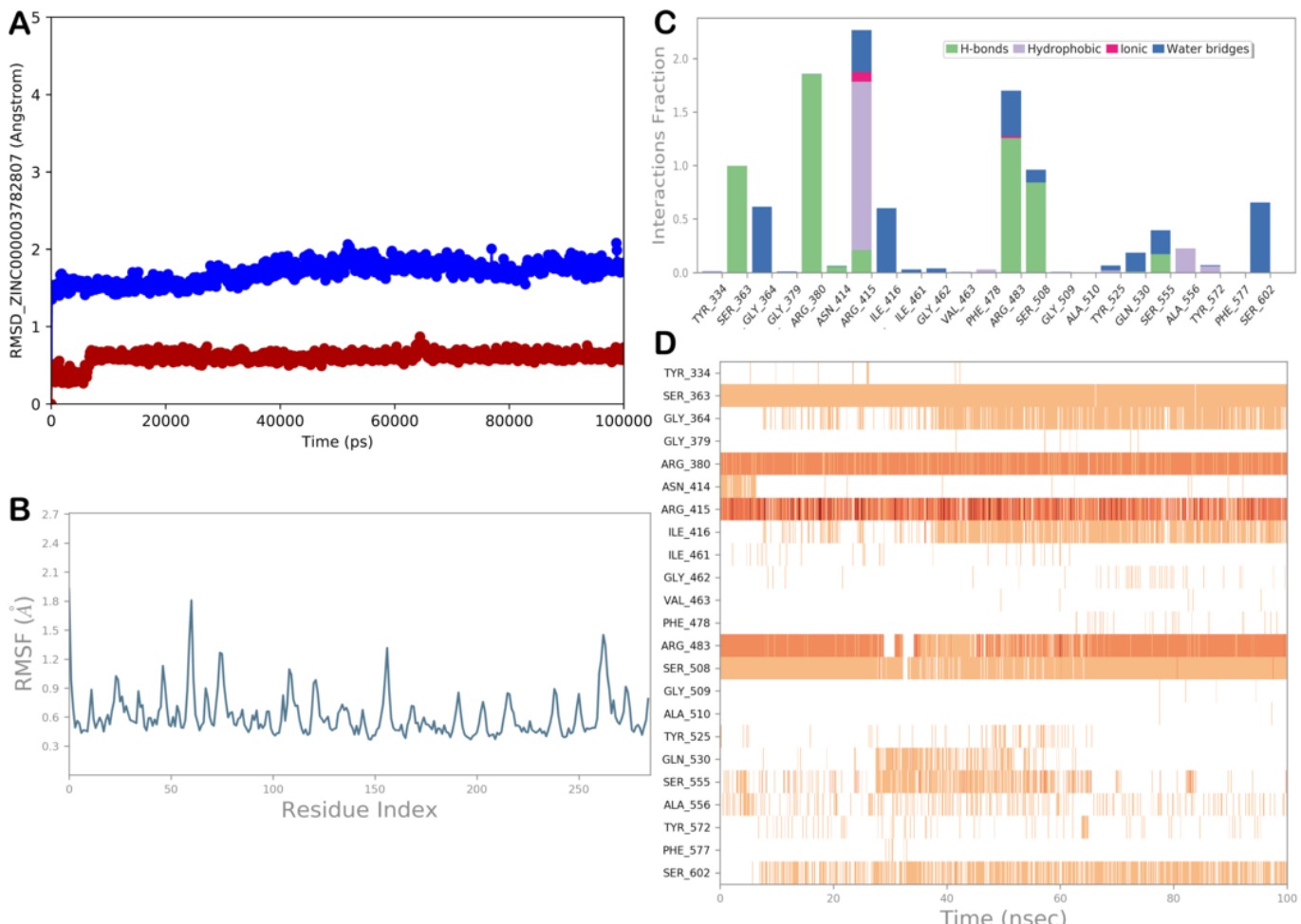

**Figure 11.** (**A**) RMSD evaluation (protein: blue line; and ligand: red line). (**B**) RMSF assessment for the complex KEAP1/ZINC000003782807, obtained by docking studies, following a 100 ns MD simulation. (**C**,**D**) ZINC000003782807 observed throughout the MD run. Four types of interactions can be distinguished: water bridges (blue), ionic (magenta), hydrophobic (grey), and H-bonds (green). Over the trajectory, the stacked bar charts are normalized. For instance, a value of 0.7 indicates that a particular contact is maintained 70% of the time during simulation. Values greater than 1.0 could occur because a protein residue could interact with the ligand more than once using the same subtype. A timeline explanation of the primary interactions is shown in the following diagram in the figure. Those residues that interact with the ligand in each trajectory frame are displayed in the output. A darker orange hue denotes several contacts that some residues have with the ligand. Maestro and Desmond software tools were utilized to generate the pictures (Maestro, Schrödinger LLC, release March 2020).

Using the complex KEAP1/ZINC000000538557, we performed an MD simulation experiment to verify the results of the molecular docking studies (Figure 13). The results of the RMSD and RMSF calculations demonstrated the strong stability of the complex with limited protein fluctuation. The MD trajectory analysis showed that the polar interactions with Arg415 and Ser508 and the hydrophobic interactions with Arg415, identified by molecular docking calculations, were preserved throughout the MD simulation, whereas the H-bond with Arg483 became sporadic. We also noticed additional interactions through hydrophobic interactions with Tyr525 and Tyr572, H-bonds with Ser555 and Ser602, and water-mediated H-bonds with the residues Ser508 and Ala556. The computational scores and this binding mode provide evidence for the capacity to target KEAP1, thereby positively modulating the NRF2 antioxidant pathway.

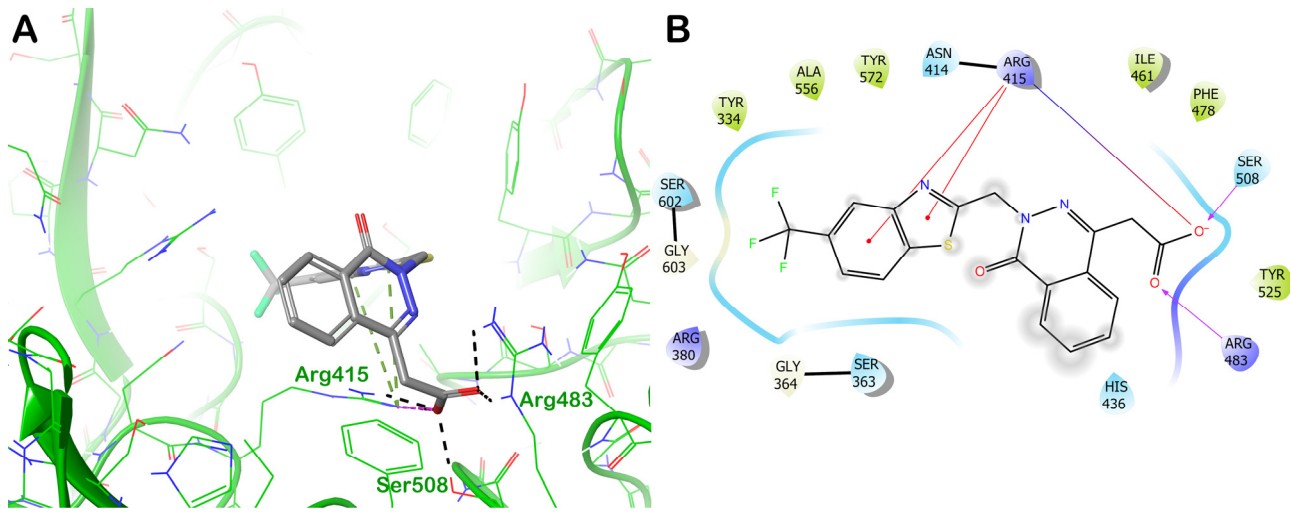

**Figure 12.** (**A**) Binding mode of ZINC000000538557 (grey sticks) within the selected KEAP1 binding site (PDB ID 2FLU, green cartoon). The interacting amino acids are represented by lines. The grey dotted lines represent the H-bonds. For the sake of clarity, non-polar hydrogens were removed. (**B**) Two-dimensional representation of the contacts established by ZINC000000538557 within the KEAP1 binding site. Maestro and Ligand Interaction Diagram applications were used for generating the pictures.

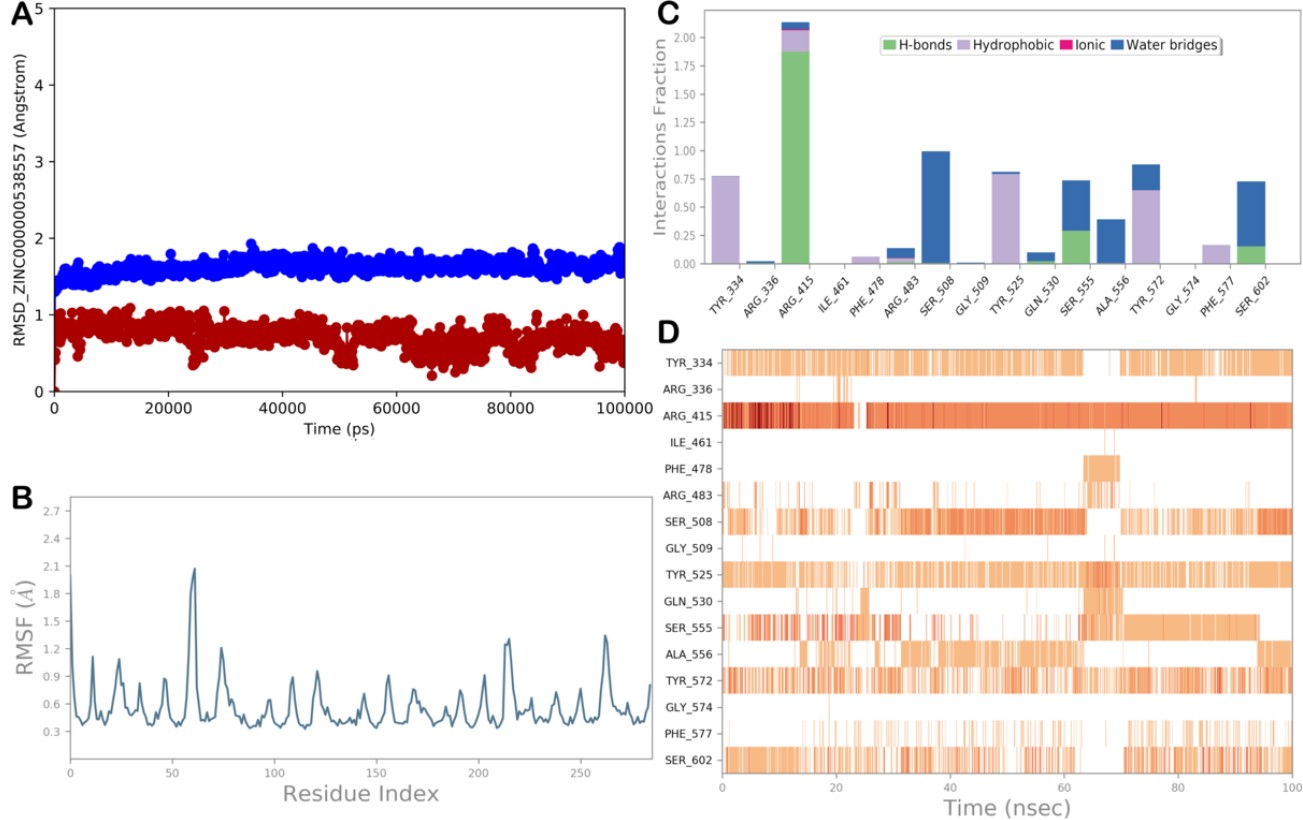

**Figure 13.** (**A**) RMSD evaluation (protein: blue line; and ligand: red line). (**B**) RMSF assessment for the complex KEAP1/ZINC000000538557, obtained by docking studies, following a 100 ns MD simulation. (**C**,**D**) ZINC000000538557 observed throughout the MD run. Four types of interactions can be distinguished: water bridges (blue), ionic (magenta), hydrophobic (grey), and H-bonds (green). Over the trajectory, the stacked bar charts are normalized. For instance, a value of 0.7 indicates that a

particular contact is maintained 70% of the time during simulation. Values greater than 1.0 could occur because a protein residue could interact with the ligand more than once using the same subtype. A timeline explanation of the primary interactions is shown in the following diagram in the figure. Those residues that interact with the ligand in each trajectory frame are displayed in the output. A darker orange hue denotes several contacts that some residues have with the ligand. Maestro and Desmond software tools were utilized to generate the pictures (Maestro, Schrödinger LLC, release March 2020).

ZINC000003948738 (Bempedoic Acid)

The third drug selected by the high-throughput docking protocol was ZINC000003948738 (bempedoic acid). This compound could establish a strong H-bond network and salt bridges with the residues Ser363, Arg380, Asn382, Arg415, Arg483, and Ser508 (Figure 14).

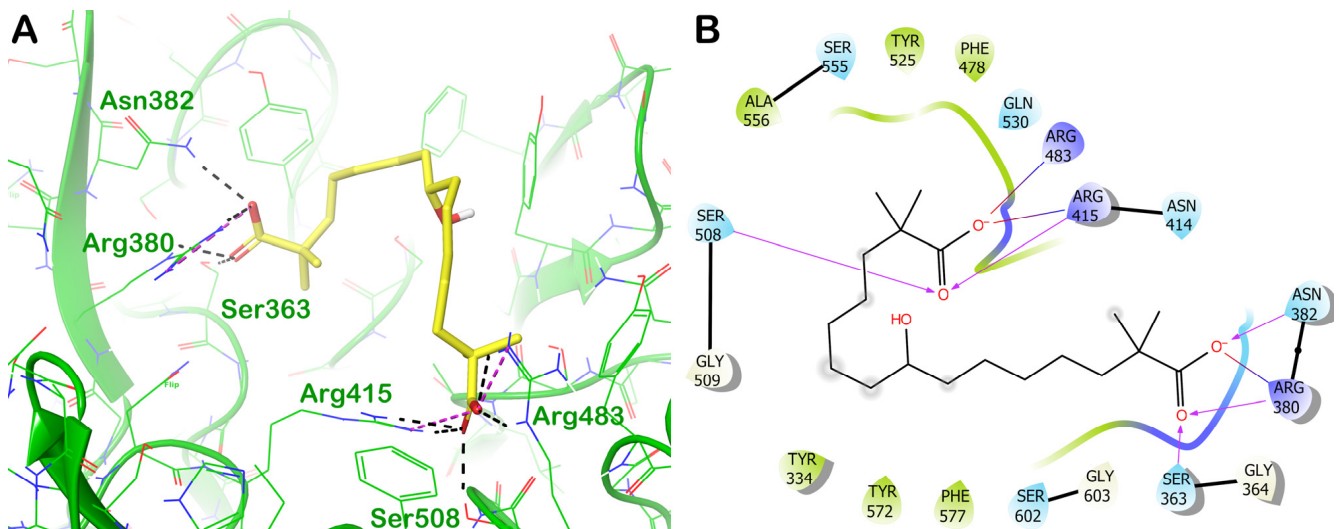

**Figure 14.** (**A**) Binding mode of ZINC000003948738 (yellow sticks) within the selected KEAP1 binding site (PDB ID 2FLU, green cartoon). The interacting amino acids are represented by lines. The grey dotted lines represent the H-bonds. For the sake of clarity, non-polar hydrogens were removed. (**B**) Two-dimensional representation of the contacts established by ZINC000003948738 within the KEAP1 binding site. Maestro and Ligand Interaction Diagram applications were used for generating the pictures.

To confirm the molecular docking results, we conducted an MD simulation study on the complex KEAP1/ZINC000003948738 (Figure 15). The calculations of the RMSD and RMSF values indicated a high stability of the complex with a limited protein fluctuation. The analysis of the MD trajectory showed that the polar contacts with the residues Ser363, Arg380, Asn382, Arg415, Arg483, and Ser508 were maintained during the MD simulation, although the H-bond with Ser363 became water-mediated. We observed further contacts with the residues Asn414 and Ser602 via H-bonds and with Tyr334 and Tyr525 via hydrophobic interactions. This binding mode, along with the computational scores, supported the ability of ZINC000003948738 to target KEAP1, potentially exerting an antioxidant profile.

In conclusion, the selected drugs, devoid of significant toxicity, could also be repurposed to promote NRF2 activation, exerting antioxidant effects through the KEAP1/NRF2 pathway, and could be considered for the prevention or treatment of OxS-induced disorders.

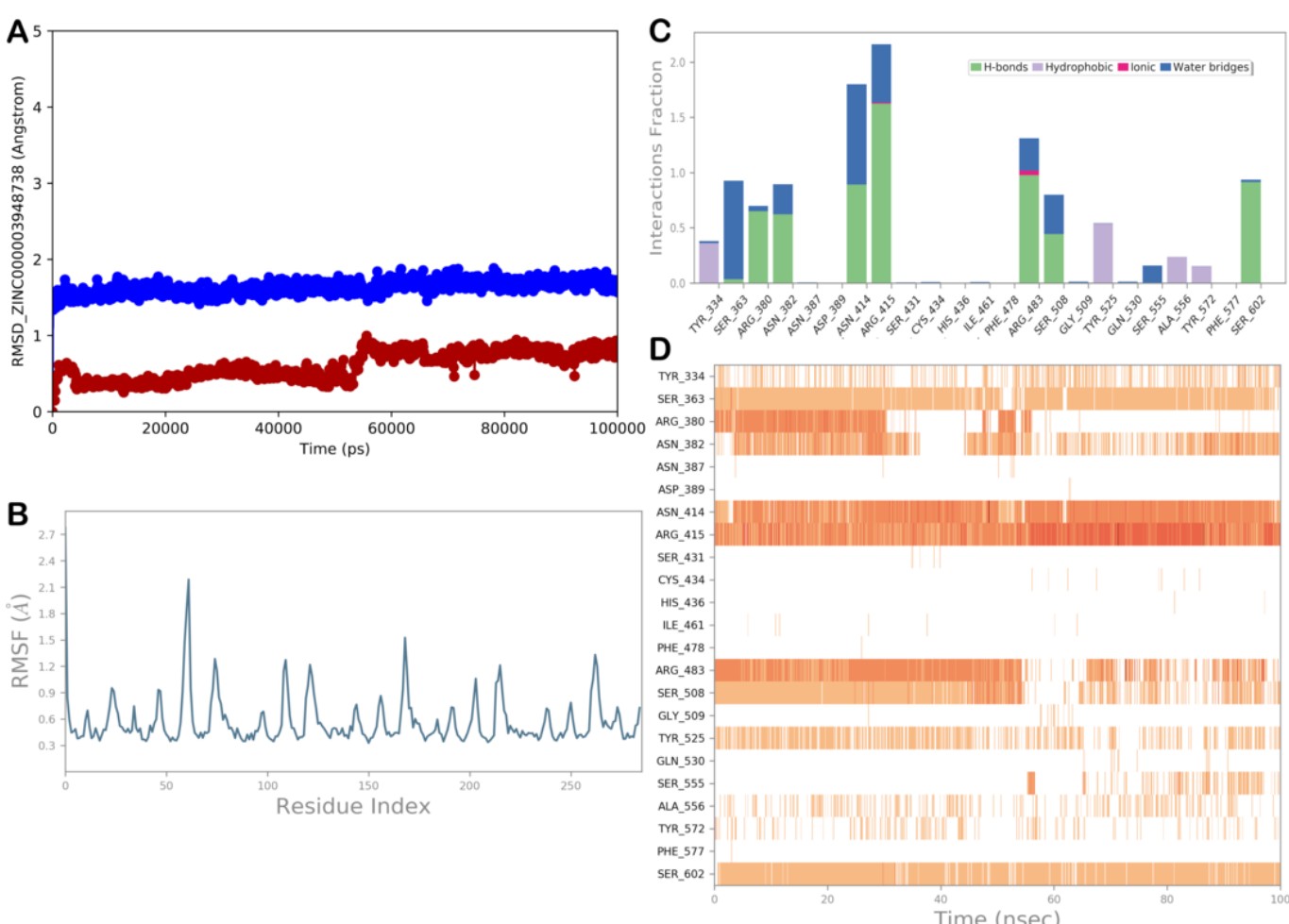

**Figure 15.** (**A**) RMSD evaluation (protein: blue line; and ligand: red line). (**B**) RMSF assessment for the complex KEAP1/ZINC000003948738, obtained by docking studies, following a 100 ns MD simulation. (**C,D**) ZINC000003948738 observed throughout the MD run. Four types of interactions can be distinguished: water bridges (blue), ionic (magenta), hydrophobic (grey), and H-bonds (green). Over the trajectory, the stacked bar charts are normalized. For instance, a value of 0.7 indicates that a particular contact is maintained 70% of the time during simulation. Values greater than 1.0 could occur because a protein residue could interact with the ligand more than once using the same subtype. A timeline explanation of the primary interactions is shown in the following diagram in the figure. Those residues that interact with the ligand in each trajectory frame are displayed in the output. A darker orange hue denotes several contacts that some residues have with the ligand. Maestro and Desmond software tools were utilized to generate the pictures (Maestro, Schrödinger LLC, release March 2020).

### 3.3. Covalent Docking Studies

It has been established that the positive regulation of the KEAP1/NRF2 pathway could also occur by covalently targeting cysteine residues that are crucial for KEAP1 functions. In particular, 27 reactive cysteine residues in the KEAP1 protein were found to be sensitive to the increase of OxS, ROSs, and electrophile agents, activating the NRF2 that became able to interact with Mafs forming a transcription complex able to target the ARE sequence, increasing the expression of antioxidant and cytoprotective genes. Accordingly, in the literature was reported that some of the cysteine residues are crucial in the regulation of KEAP1 functions, including the recognition of NRF2 and the subsequent activation of the mentioned protein, as well as the negative modulation of client proteins such as Cul-3. In fact, this protein, along with RBX1, can form a complex with Cul-3/RBX1, which is necessary for

the ubiquitination process performed by the E3 ligase after the interaction of the NRF2 with the KEAP1. Thus, NRF2 can be degraded by proteasome 26S. The lack of formation of the Cul-3/RBX1 complex mediated by the KEAP1 results in the absence of NRF2 degradation, which can be activated by the phosphorylation process to enter the nucleus, form a complex with Mafs, and participate in the transcription of antioxidant and cytoprotective genes. Moreover, at least eight cysteine residues, namely Cys77, Cys151, Cys257, Cys273, Cys288, Cys297, Cys434, and Cys613, are directly involved in redox sensing and NRF2 activation. In this context, several synthetic small molecules have been investigated, including approved drugs (e.g., dimethyl fumarate, an approved drug for treating relapsing multiple sclerosis), for their ability to form covalent adducts with regulatory cysteine residues. More interesting is the investigation of natural products acting as covalent KEAP1 binders. Because of their reduced toxicity and undesired effects, natural products can be administered as food supplements and/or included in the dietary regimen. Some interesting examples of this class of compounds capable of modifying cysteine residues on KEAP1 have been reported in the literature. In particular, several reports have identified natural products involved in the formation of covalent adducts with different cysteine residues located in different KEAP1 domains. Interestingly, one of the most sensitive cysteine residues, Cys151, was found in the BTB domain, and its function as the key regulatory amino acid for activating NRF2 was well established. Among the described natural products able to form covalent adducts with the KEAP1 protein, one of the most interesting was curcumin. This molecule can target different cysteine residues located in the regulatory domains of KEAP1, thereby interfering with the interaction with the NRF2. The yellow pigment belonging to *Curcuma longa* possesses a plethora of significant pharmacological properties, including antioxidant, antimicrobial, anti-inflammatory, anticancer, neuroprotective, and anti-ischemic effects. Regarding the antioxidant profile, several studies have highlighted that curcumin exerts this effect by acting on the regulatory cysteine residues of KEAP1. The chemical structure of curcumin showed two Michael acceptor functions belonging to the $\alpha,\beta$-unsaturated 1,3-diketone moiety. These reactive groups were able to target cysteine residues considering the reaction that can occur between the nucleophilic moiety of cysteine (-SH group) and the unsaturated ketone function. As previously mentioned, several cysteine residues of KEAP1 are susceptible to the formation of covalent adducts. Regarding curcumin, Shin et al. investigated the preferred cysteine binding of curcumin to establish its possible mechanism of action. Their findings clearly indicated that Cys151 is required for NRF2 transcriptional activity in response to curcumin treatment, thus exerting its antioxidant effect [88]. Although curcumin was found to form covalent adducts with Cys273 and Cys288, mutagenesis studies clearly indicated that Cys151 was the most reactive against curcumin. Accordingly, based on the structural analysis of our top-ranked natural products found to potentially target the Kelch 1 domain, we found that all three natural compounds contained functional groups that can be relevant in forming covalent adducts with the susceptible residues. In fact, ZINC000059204232 (isoxanthochymol) presented two possible functional groups able to establish covalent adducts: a carbonyl group and unsaturated carbon atoms. ZINC000001531844 (gingerenone A), due to the structural similarity with curcumin, shares one $\alpha,\beta$-unsaturated diketone moiety, and ZINC000000338310 (meranzin hydrate) included in its structure a carbonyl group that could be involved in forming covalent adducts with the susceptible residues. To investigate the possible capability of these compounds to form covalent adducts with Cys151 of the KEAP1 protein, we applied a covalent docking protocol to assess whether the covalent reactions were energetically favorable for the mentioned compounds. The output of the covalent docking calculation is shown in Figure 16.

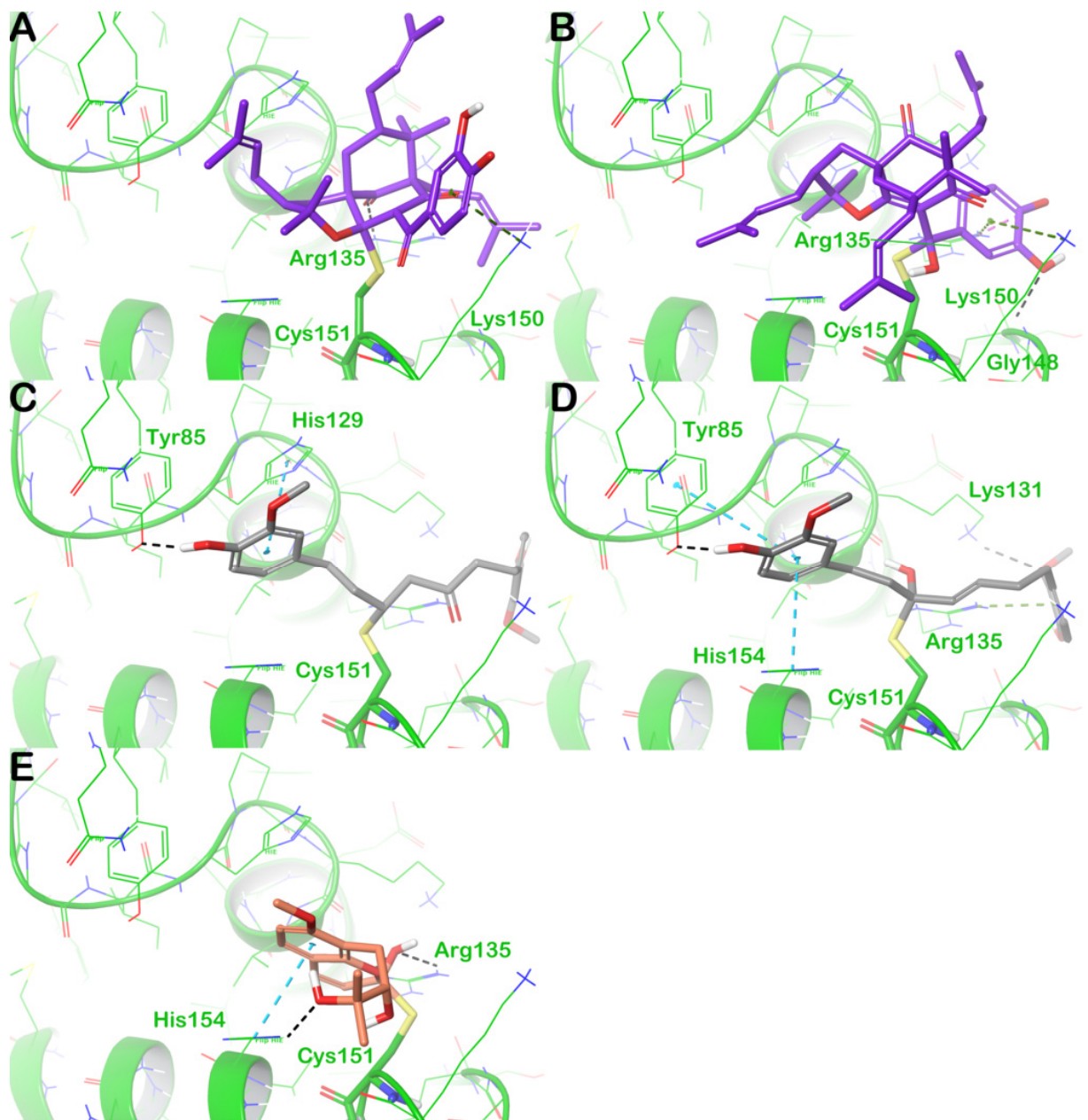

**Figure 16.** Covalent docking results considering Cys151 located in the BTB domain of KEAP1 (green cartoon, PDB ID 7EXI): (**A,B**) ZINC000059204232 (isoxanthochymol); (**C,D**) ZINC000001531844 (gingerenone A); and (**E**) ZINC000000338310 (meranzin hydrate). The reactive Cys151 is represented by sticks, while the key interacting residues are represented by lines. Pictures were generated by Maestro and Desmond software tools (Maestro, Schrödinger LLC, release March 2020).

Starting from the compound ZINC000059204232 (isoxanthochymol), two possible functional groups able to establish covalent adducts were susceptible to the attack of Cys151. However, based on the favorable energies required to form the covalent adduct, both reactions can occur, as shown in Figure 16A (a CovDock score double bond of −2.827 kcal/mol) and in Figure 16B (a CovDock score of −1.982 kcal/mol). For both potential binding modes, further contact with the residues Arg135 and Lys150 could contribute to stabilizing the covalent adduct formed by the selected compound. Next, the compound ZINC000001531844 (gingerenone A) was also able to form a covalent adduct by its functional groups that were able to participate in this type of reaction. In particular, the compound ZINC000001531844 could covalently target Cys151 via the Michael acceptor and the carbonyl group in its

structure. Based on the CovDock scores, the reaction involving the Michael acceptor (Figure 16C) was the most favorable (a CovDock score of −4.071 kcal/mol) with respect to that involving the carbonyl group (a CovDock score of −3.454 kcal/mol) (Figure 16D). In addition, in this case, further contact with the residues Tyr85 and His129 could contribute to the stabilization of the adduct when the Michael acceptor is involved in the covalent bond, whereas the residues Tyr85, Lys131, Arg135, and His154 could stabilize the adduct formed when the carbonyl group is involved in the covalent reaction. Finally, the natural product ZINC000000338310 (meranzin hydrate) can covalently target the residue Cys151 by its carbonyl group, allowing the formation of an energetically favorable adduct (a CovDock score of −3.924 kcal/mol); furthermore, contacts with Arg135 and His154 stabilized the formed covalent complex (Figure 16E).

Accordingly, based on the computational evaluation, the selected natural products could present a potential dual mechanism in modulating KEAP1 functions by targeting both the Kelch 1 domain, establishing non-covalent interactions within the NRF2 interacting site, and the covalently bound Cys151 in the BTB domain. Remarkably, this dual mechanism can improve the ability to inhibit the formation of the KEAP1/NRF2 complex, producing a longer activation of NRF2 and consequently a stronger antioxidant effect.

## 4. Conclusions

In summary, we have presented the application of a computer-based protocol for screening chemical libraries to identify possible antioxidant agents targeting the KEAP1/NRF2 pathway. In particular, considering the NRF2 binding site on the KEAP1, we developed a computational screening aimed at identifying compounds that could directly target the Kelch 1 domain, which physically hampered the recognition of KEAP1 and NRF2. In addition, we evaluated the capability of some retrieved compounds to act as covalent ligands targeting the reactive cysteine residue Cys151. Specifically, screening was performed using two properly prepared chemical libraries, one containing natural products and the other containing world-approved and investigational drugs. After computer-based filtering criteria and submission of the resulting complexes to MD simulation experiments, we identified a few suitable compounds potentially able to target KEAP1 and to activate the NRF2 pathway, thereby exerting an antioxidant effect. Considering the natural products, isoxanthochymol, gingerenone A, and meranzin hydrate showed the best predicted profile for behaving as antioxidant agents, whereas, among the drugs, nedocromil, zopolrestat, and bempedoic acid could be considered for a repurposing approach to identify possible antioxidant agents. Regarding world-approved and investigational drugs, several preclinical and clinical studies have already been conducted for their therapeutic indications. These molecules show excellent stability and pharmacokinetic profile, have mild common side effects, and have good economic viability. Specifically, as reported in Table 2, nedocromil is a drug approved for the treatment of asthma with an anti-inflammatory action that has a direct application to the bronchial mucosa through inhalation or as an ophthalmic formulation [89]. Zopolrestat is an investigational drug that is not yet on the market as a brand but that has undergone a series of preclinical investigations for the treatment of diabetic complications, such as diabetic cardiovascular autonomic neuropathy or diabetic neuropathy [90]. Bempedoic acid, approved for the treatment of hypercholesterolemia, is available in the market both alone and in combination with ezetimibe. It has an excellent ADMET profile and increases the blood levels of statins, especially simvastatin and pravastatin [91,92]. Interestingly, because of the limited toxicity of the mentioned drugs (Table 2), they may be used for a long period to promote antioxidant effects. Obviously, the dose regimen for exerting antioxidant effects must be evaluated in preclinical studies to provide a potential translation from the clinical setting. In particular, we would like to highlight that several studies have demonstrated that the inhalation route for some natural products and drugs (i.e., resveratrol, vitamin E, budenoside, and hexamethonium) could be a viable way to positively act on diseases involving OxS [93,94]. To this end, the drug

nedocromil was already formulated for inhalation administration and could be investigated as an antioxidant inhalation agent.

Remarkably, we hypothesized a possible mechanism of action of these compounds for their possible antioxidant activity by targeting the KEAP1/NRF2 pathway, which to the best of our knowledge has not been previously described. Accordingly, the identified compounds could represent a significant starting point for the discovery of novel and effective antioxidant agents that are devoid of potential toxicity issues and can be administered for long periods to exert antioxidant effects and to prevent and/or treat several diseases in which OxS is crucial for their development and progression. Further biological studies will clarify the role of these molecules, indicating which could be effective antioxidant agents.

**Author Contributions:** Conceptualization, S.B. and V.C.; methodology, S.B., I.G., L.F. and H.S.; software, S.B., I.G., L.F. and H.S.; validation, S.B., I.G., L.F., H.S. and V.C.; formal analysis, S.B., I.G., L.F., H.S. and V.C.; investigation, S.B., I.G., L.F. and H.S.; data curation, S.B. and I.G.; writing—original draft preparation, S.B.; writing—review and editing, S.B., I.G., L.F., H.S. and V.C. All authors have read and agreed to the published version of the manuscript.

**Funding:** This research received no external funding.

**Data Availability Statement:** Data are contained within the article.

**Conflicts of Interest:** The authors declare no conflict of interest.

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
