# Peer review of "In Silico Identification of Natural Products and World-Approved Drugs Targeting the KEAP1/NRF2 Pathway Endowed with Potential Antioxidant Profile"

_computation, doi:10.3390/computation11120255_

Round 1
Reviewer 1 Report
Comments and Suggestions for Authors
This manuscript by Brogi et al. aimed to select novel antioxidant agents to reduce oxidative stress via a variety of computational methods. They considered the NRF2 binding site on KEAP1 protein to conduct ligand screening of two compound libraries (natural products and world-approved drugs). Several potential compounds were hit by the evaluation of binding energy and ADMET properties, and their binding stability was further verified by MD simulation. The identified compounds represent a significant starting point for the discovery of potential antioxidant agents. The manuscript was well written and presented in a logical way. This reviewer recommended its publication in the journal of Computation after addressing the following comments.
(1) This reviewer prefers not to use "we present a computer-based protocol" in the Abstract (Line 11) and "we have developed a computer-based protocol" in the Introduction part (Line 65). The protocols used in this manuscript are not special or developed by the authors themselves, and they appeared to be standard ones and are used in many publications.
(2) The words "FDA-approved drugs" mentioned in the title seemed to be inaccurate. In fact, as stated in the main text, the compound library used for screening consists of natural products and world-approved drugs.
(3) Which structure was actually used as the receptor in the docking (Lines 108-109)? Is it 2FLU or 7EXI?
(4) Section 2.4: The authors used the OPLS force field in combination with TIP3P water model. Why choose TIP3P? In general, the OPLS force field should be used with the native TIP4P water model.
(5) The GlideScore given in Table 1 is for the SP score or XP score?
(6) How did the interaction fraction in Figure 4C was defined and calculated?
Author Response
Reviewer 1
this manuscript by Brogi et al. aimed to select novel antioxidant agents to reduce oxidative stress via a variety of computational methods. They considered the NRF2 binding site on KEAP1 protein to conduct ligand screening of two compound libraries (natural products and world-approved drugs). Several potential compounds were hit by the evaluation of binding energy and ADMET properties, and their binding stability was further verified by MD simulation. The identified compounds represent a significant starting point for the discovery of potential antioxidant agents. The manuscript was well written and presented in a logical way. This reviewer recommended its publication in the journal of Computation after addressing the following comments.
Authors: We sincerely thank the reviewer for the positive evaluation of our manuscript. All the points have been addressed in the revised manuscript.
(1) This reviewer prefers not to use "we present a computer-based protocol" in the Abstract (Line 11) and "we have developed a computer-based protocol" in the Introduction part (Line 65). The protocols used in this manuscript are not special or developed by the authors themselves, and they appeared to be standard ones and are used in many publications.
Authors: According to the reviewer’s suggestion, we have removed the sentences, and replaced the terms with we have applied a computer-based protocol.
(2) The words "FDA-approved drugs" mentioned in the title seemed to be inaccurate. In fact, as stated in the main text, the compound library used for screening consists of natural products and world-approved drugs.
Authors: Thank you for your pertinent observation. Accordingly, in the title, we have replaced the term FDA-approved drugs with World-approved drugs.
(3) Which structure was actually used as the receptor in the docking (Lines 108-109)? Is it 2FLU or 7EXI?
Authors: We have used both structures. In particular, screening was performed using 2FLU because it is the Kelch1 domain with the interacting region of the Nrf2 protein, whereas 7EXI is the BTB domain of Keap1 and was used for covalent docking because the reactive cysteine Cys151 is enclosed in that domain. The 7EXI crystal structure was used only for covalent docking, taking into account the natural products containing features that could allow covalent binding to the mentioned residue because they present features already found in natural products that covalently bind Keap1 in the BTB domain. Accordingly, we have introduced this clarification in the Materials and Methods section to avoid any mistake.
(4) Section 2.4: The authors used the OPLS force field in combination with TIP3P water model. Why choose TIP3P? In general, the OPLS force field should be used with the native TIP4P water model.
Authors: According to Desmond user manual and a very large number of published articles, we observed that OPLS force field is applicable to liquid system, mainly applied to polypeptide, protein, nucleic acid, organic solvent and other liquid system, generally applicable to the water model of TIP3P or TIP4P, and in fact, OPLS simulations in aqueous solution typically use the mentioned water models (10.1186/s40824-023-00386-7; Desmond user manual). Furthermore, the TIP3P model is compatible with the OPLS parameterization (10.1063/1.445869; 10.1021/acs.jctc.8b01039). Using OPLS-AA force field and in particular OPLS3 in combination with TIP3P model, it is possible to obtain reliable results, saving some time. Usually, we use this combination of force field and water model and we experienced from about 15 years that is suitable in virtual screening, rational drug design, long-time simulation to study biological systems and med chem purpose (some examples: 10.1021/jm301370e; 10.1016/j.compbiomed.2021.104808; 10.3390/v15122291; 10.3390/molecules28186535; 10.3390/pathogens11091020; 10.1007/s00894-022-05270-0; 10.1039/D0FO01511C; 10.3390/computation10010007; 10.3390/computation10050069; 10.3389/fchem.2019.00574). Moreover, in the literature this such combination is vastly used in molecular modelling and computational chemistry (10.1074/jbc.M117.791467; 10.1007/s10822-016-9974-4; 10.1038/s41598-018-31080-7; 10.1002/bkcs.11382; 10.1371/journal.pcbi.1005314; 10.1021/cs401047k; 10.1039/C1CP20323A; 10.1073/pnas.0408037102; 10.1186/s43141-023-00621-7; 10.1016/j.bpj.2015.07.018 only for citing some examples).
(5) The GlideScore given in Table 1 is for the SP score or XP score?
Authors: The values of GlideScore are referred to SP and are reported in Table 1 and Table 2 (we added this detail in the Tables). We used SP for screening and evaluated the XP score to assess the reliability of the poses in terms of docking scores and binding mode. As reported in the Materials and Methods section we wrote “we selected compounds with SP and XP scores < −8.00 kcal/mol for the natural products, and < −6.5 kcal/mol for the drugs, and with comparable binding modes derived from the different scoring functions.”
(6) How did the interaction fraction in Figure 4C was defined and calculated?
Authors: We apologize for the lack of details regarding the interaction fraction and the evaluation of the number and quality of contacts established by the ligand within the selected binding site during the MD simulation. The types of interactions (Figures 4, 6, 8, 11, 13, and 15 panel C) can be categorized by type and are summarized as follows: hydrogen bonds, hydrophobic, ionic, and water bridges. Each interaction type contains more specific subtypes, which can be explored through the Simulation Interactions Diagram panel. Stacked bar charts are normalized over the course of the trajectory. For example, a value of 0.7 of interaction fraction suggests that the specific interaction is maintained for 70% of the simulation time. Values over 1.0 are possible because some protein residues may make multiple contacts of the same subtype with the ligand. The amendment of the mentioned Figure captions was done.
Reviewer 2 Report
Comments and Suggestions for Authors
What would be "chemical libraries properly curated" ?
The abstract needs improvement in the results section. No concentration results have been reported in this topic.
I find the introduction very sparse. Long paragraphs without references. For example, 7 lines of text for 1 reference, then 14 lines of text for 2 references. I suggest reorganizing the introduction, with a better literature search.
The PDB code of the structures used was missing.
The amino acids in the 2D images are difficult to see.
It would be interesting if, before the conclusion, the authors mentioned how they believe these structures might behave in in vivo studies. What they expect from a practical study, economic viability, long-term stability of the structures and acceptance in humans.
Author Response
Reviewer2
What would be "chemical libraries properly curated" ?
Authors: We have used this term considering that each library used was prepared ex novo, in fact considering the natural products we enclosed in the library several libraries containing natural products as reported in the Materials and Methods section (AfroDb (African medicinal plant), FoodDb, TCM NP (Traditional Chinese medicine), HDMB plant, TIMTEC NP, MolPort NP, IB Screen NP, NPACT, HITNP (herbal ingredients), HIMNP (herbal ingredients in vivo metabolism), Aster NP, SPECS NP, NUBBE natural products, NUTRICHEM, DrugBank Nutraceuticals, Prestwick Phyto NP, DNP natural products, UEFS natural products, TargetMol NP, and INDOFINE natural products). Accordingly, we removed redundant structures, and the resulting compounds were prepared as described in section 2.1. Same for the drugs. In fact, we downloaded some databases to generate a unique library containing approved drugs. In any case, we removed the term properly curated with properly prepared.
The abstract needs improvement in the results section. No concentration results have been reported in this topic.
Authors: We thank the reviewer for the careful reading. According to this comment, we have improved the abstract by reporting details on the results obtained and information on the identified drugs to be employed as antioxidant agents.
I find the introduction very sparse. Long paragraphs without references. For example, 7 lines of text for 1 reference, then 14 lines of text for 2 references. I suggest reorganizing the introduction, with a better literature search.
Authors: We thank the reviewer for this suggestion. Accordingly, the introduction was completely rewritten to improve the content, including the number of references, highlighting the role of OxS and ROS in different pathophysiological processes, the importance of antioxidant agents, and the significance of targeting the Keap1/Nrf2 pathway.
The PDB code of the structures used was missing.
Authors: The PDB code of the structures used has been reported in the Materials and Methods section 2.1 “The three-dimensional structure of KEAP1 with the NRF2 interacting region was downloaded from the protein data bank (PDB ID 2FLU). In addition, we used the crystal structure of the BTB domain of KEAP1 (PDB ID 7EXI) (PDB, https://www.rcsb.org/ accessed on 21 April 2023).” Furthermore, this information was also added in section 2.5 Covalent docking. Accordingly, for each Figure, in the caption was reported the corresponding structure that was used.
The amino acids in the 2D images are difficult to see.
Authors: to help readers to see the amino acids in the ligand-interaction diagram, we introduced novel Figures with larger font sizes to guarantee better reading.
It would be interesting if, before the conclusion, the authors mentioned how they believe these structures might behave in in vivo studies. What they expect from a practical study, economic viability, long-term stability of the structures and acceptance in humans.
Authors: We thank the reviewer for this constructive comment. Accordingly, we expanded the previous discussion on natural products and drugs to address the points raised by the reviewer. In particular, in the conclusion section, when we summarized the findings considering the drugs, we have reported a series of details on their preclinical evaluation. These molecules show excellent stability and pharmacokinetic profile, have common mild side effects, and have good economic viability. Moreover, because of their limited toxicity, it is possible to administer them for a long time to exert antioxidant effects. Obviously, the dose regimen for exerting antioxidant effects must be evaluated in preclinical studies to provide a potential translation in the clinical setting. In particular, we would like to highlight that several studies have demonstrated that the inhalation route for some natural products and drugs (i.e., resveratrol, vitamin E, budenoside, and hexamethonium) could be a viable way to positively act on diseases involving OxS. To this end, for example, the drug nedocromil was already formulated for inhalation administration and could be investigated as an antioxidant inhalation agent.
Round 2
Reviewer 1 Report
Comments and Suggestions for Authors
The authors have addressed all of my concerns in the revised manuscript.